# Phylogenomics reveals the evolutionary origins of lichenization in chlorophyte algae

Camille Puginier [1], Cyril Libourel [1], Juergen Otte[2], Pavel Skaloud [3], Mireille Haon[4,5], Sacha Grisel [4,5], Malte Petersen [6], Jean-Guy Berrin [4,5], Pierre-Marc Delaux [1] ✉, Francesco Dal Grande [2,7,8] ✉ & Jean Keller [1,9] ✉

Mutualistic symbioses have contributed to major transitions in the evolution of life. Here, we investigate the evolutionary history and the molecular innovations at the origin of lichens, which are a symbiosis established between fungi and green algae or cyanobacteria. We de novo sequence the genomes or transcriptomes of 12 lichen algal symbiont (LAS) and closely related non-symbiotic algae (NSA) to improve the genomic coverage of Chlorophyte algae. We then perform ancestral state reconstruction and comparative phylogenomics. We identify at least three independent gains of the ability to engage in the lichen symbiosis, one in Trebouxiophyceae and two in Ulvophyceae, confirming the convergent evolution of the lichen symbioses. A carbohydrate-active enzyme from the glycoside hydrolase 8 (GH8) family was identified as a top candidate for the molecular-mechanism underlying lichen symbiosis in Trebouxiophyceae. This GH8 was acquired in lichenizing Trebouxiophyceae by horizontal gene transfer, concomitantly with the ability to associate with lichens fungal symbionts (LFS) and is able to degrade polysaccharides found in the cell wall of LFS. These findings indicate that a combination of gene family expansion and horizontal gene transfer provided the basis for lichenization to evolve in chlorophyte algae.

Mutualistic interactions between plants and microorganisms are the foundation of plant diversification and adaptation to almost all terrestrial ecosystems[1,2]. An emblematic example of mutualism impact on Earth is the transition of plants from the aquatic environment to land, which occurred 450 million years ago and was partly enabled by the arbuscular mycorrhizal symbiosis formed with Glomeromycota fungi[2,3]. Another emblematic example of a plant-fungi symbiosis occurs in the mutualistic association between certain chlorophyte algae and fungi resulting in the formation of lichens[4,5].

Lichens are symbiotic structures composed of several types of organisms including a fungal partner, that most commonly belongs to the Ascomycetes and more rarely to the Basidiomycetes, and a photosynthetic partner, also called photobiont. Photobionts can be either cyanobacteria or algae belonging to the Chlorophytes. Certain lichens can contain both types of photobionts[6]. In this mutualistic symbiosis, both mycobionts and photobionts obtain benefits from their association. Carbohydrates from photosynthesis are supplied to the fungal partners, whereas the fungi create a favorable microenvironment

[1]Laboratoire de Recherche en Sciences Végétales (LRSV), Université de Toulouse, CNRS, UPS, INP, Toulouse 31320 Castanet-Tolosan, France. [2]Senckenberg Biodiversity and Climate Research Centre (SBiK-F), Senckenberganlage 25, 60325 Frankfurt am Main, Germany. [3]Department of Botany, Faculty of Science, Charles University, Benátská 2, CZ-12800 Praha 2, Czech Republic. [4]INRAE, Aix Marseille Université, UMR1163 Biodiversité et Biotechnologie Fongiques (BBF), 13009 Marseille, France. [5]INRAE, Aix Marseille Université, 3PE Platform, 13009 Marseille, France. [6]High Performance Computing & Analytics Lab, University of Bonn, Friedrich-Hirzebruch-Allee 8, 53115 Bonn, Germany. [7]LOEWE Centre for Translational Biodiversity Genomics (TBG), Senckenberganlage 25, 60325 Frankfurt am Main, Germany. [8]Department of Biology, University of Padova, Padua, Italy. [9]Department of Insect Symbiosis, Max Planck Institute for Chemical Ecology, 07745 Jena, Germany. ✉e-mail: pierre-marc.delaux@cnrs.fr; francesco.dalgrande@unipd.it; jean.keller@cnrs.fr

shielding the photobionts from biotic and abiotic stresses[5,7]. Recently, metagenomic studies have demonstrated that other types of microorganisms, such as lichenicolous fungi and bacteria, are found within the lichen thallus and are likely important for its biology[8–10].

Because of their ecological and physiological importance, how these mutualistic interactions originated has been a central question for decades[11]. The comparison of genomes in a defined phylogenetic context (comparative phylogenomics) has successfully unraveled the evolutionary history of several mutualistic symbioses with complex evolutionary patterns, combining gains and losses across lineages[12–14]. In addition, such approaches have the potential to identify the molecular mechanisms associated with major innovations, including symbioses[15–17]. Even though lichens have been considered as a long-lasting mutualistic interaction between lichen fungal symbionts (LFS) and one or more photobionts, lichens have been asymmetrically investigated from the fungal perspective leading to the conclusion that the ability to form lichens has been originally acquired, lost, and regained multiple times during the evolution of the ascomycetes and basidiomycetes[18–20].

On the photobiont side, algal species that are known to establish the lichen symbioses (thereafter called lichen algal symbionts or LAS) are almost exclusively found in two of the eleven chlorophyte algae classes, the Ulvophyceae and the Trebouxiophyceae[5,21]. Such distribution of the ability to associate with LFS might be the result of either a single gain in the common ancestor of Ulvophyceae and Trebouxiophyceae followed by multiple losses, in a similar manner to other terrestrial endosymbioses[2,12,13,22], or multiple independent gains. Studies based on time-calibrated phylogenetic approaches provided strong support for the convergent evolution of LFS and suggested a similar pattern for LAS[20,23]. However, the limited availability of LAS genomes has so far constrained molecular analyses to single algal species such as *Asterochloris glomerata* and *Trebouxia sp.* TZW2008[24,25]. Thus, the evolutionary history of lichens on the green algal side and the underlying molecular mechanisms associated with lichenization (algae that are hosted and have a lifestyle inside of lichens symbioses) remain elusive. The initiation of contact between lichen symbionts hinges on mutual recognition, with emerging evidence suggesting the involvement of elicitors that interact with the cell wall (reviewed in[26]). On the mycobiont side, fungal stimuli may encompass the activities of carbohydrate-active enzymes (CAZymes), potentially enhancing the permeability of algal cell walls[27]. Sugars, sugar alcohols, along with other compound groups like secondary metabolites and antioxidants, are proposed as key elements in maintaining the

intricately balanced symbiotic interplay between fungi and algae in lichens[26,28]. This process implies that LAS should manifest distinct genomic features compared to algae unable to establish symbiotic associations[29–32]. In this case as well, the limited availability of genomic information for LAS has thus far impeded the testing of this hypothesis.

In this study, we deployed unsupervised phylogenomic comparative approaches to decipher the evolutionary history and the genetic mechanisms conferring certain chlorophyte species the ability to engage in the lichen symbiosis. In this study, we de novo sequenced and annotated six LAS genomes, two LAS transcriptomes, three non-symbiotic algae (NSA) genomes, and one NSA transcriptome. We performed ancestral state reconstruction using this dataset along with 26 genomes and 103 transcriptomes of chlorophyte algae publicly available, demonstrating at least three convergent gains of lichenization in chlorophyte algae. We scrutinized one of these events through comparative phylogenomics cross-referenced with differential gene expression data and identified lichenization-related molecular mechanisms. We propose an evolutionary model for the evolution of lichens based on the projection of these molecular characteristics onto the phylogeny of chlorophyte algae. This scenario involves the expansion of gene families and horizontal gene transfers that likely facilitated the interaction between the symbiotic partners.

## Results

### Expanding the genomic coverage of the chlorophyte algae

To date, genomes and transcriptomes are available for only seven LAS. To investigate the evolution of lichens, we produced six new long-reads-based genome assemblies for LAS species belonging to the Trebouxiales, Botryococcus and Apatococcus clades (Table 1, Fig. 1, Supplementary Data 1). We also sequenced three closely related NSA including species from the Apatoccocus and Myrmecia genera for which no genomes were previously available (Table 1, Fig. 1, Supplementary Data 1). Assemblies for eight of the nine species displayed an average scaffold length N50 of almost 2 Mb, and an average of only 143 scaffolds (Table 1, Supplementary Data 2). The ninth assembly (*Apatococcus fuscideae* SAG2523) displayed a scaffold N50 of 50 kb and a much higher number of 2,319 scaffolds (Table 1, Supplementary Data 2). However, the genome completeness, estimated by BUSCO, indicated that most of the actual proteome was captured (75.3%, Table 1). To complete this dataset, the transcriptome of three additional species, two Trebouxiophyceae, and the symbiotic Ulvophyceae *Paulbroadya petersii*, were sequenced on an Illumina NovaSeq

**Table 1 | List of species sequenced, the class they belong to, their symbiotic status (LAS: lichen algal symbionts, NSA: non-symbiotic algae), the resource type (G: genomic, T: transcriptomic), the genome sizes, the N50, the number of protein and the BUSCO completeness**

| Species | Lichens | Class | Resource type | Genome size (Mb) | N50 | Number of CDS/proteins | Busco score (%) |
|---|---|---|---|---|---|---|---|
| *Apatococcus fuscideae (ApafusSAG2523)* | LAS | Trebouxiophyceae | G | 102.683555 | 10579 | 12,399 | 75.3 |
| *Apatococcus lobatus (ApalobSAG2145)* | NSA | Trebouxiophyceae | G | 106.452463 | 15974 | 11,112 | 95.6 |
| *Coccomyxa pringsheimii (CocpriSAG2167)* | LAS | Trebouxiophyceae | G | 50.915843 | 15647 | 10,022 | 95.4 |
| *Elliptochloris bilobata (EllbilSAG24580)* | LAS | Trebouxiophyceae | G | 52.254682 | 9434 | 8676 | 93.7 |
| *Myrmecia biatorellae (MyrbiaSAG882)* | LAS | Trebouxiophyceae | T | NA | NA | 15,547 | 73.6 |
| *Myrmecia bisecta (MyrbisSAG2043)* | NSA | Trebouxiophyceae | G | 83.484054 | 15027 | 12,551 | 96.9 |
| *Symbiochloris irregularis (SymirrSAG2036)* | NSA | Trebouxiophyceae | G | 65.271117 | 10287 | 10,921 | 91 |
| *Trebouxia sp (TrespOTU5)* | LAS | Trebouxiophyceae | G | 68.275875 | 70602 | 11,710 | 94.4 |
| *Trebouxia sp (TrespOTU1)* | LAS | Trebouxiophyceae | G | 70.863246 | 9004 | 12,712 | 96.3 |
| *Trebouxia sp (TrespOTU3)* | LAS | Trebouxiophyceae | G | 62.215734 | 9923 | 11,096 | 93 |
| *Paulbroadya petersii (PaupetSAG2240)* | LAS | Ulvophyceae | T | NA | NA | 18,999 | 76.8 |
| *Paulbroadya prostrata (PauproSAG2392)* | NSA | Ulvophyceae | T | NA | NA | 15,610 | 81.9 |

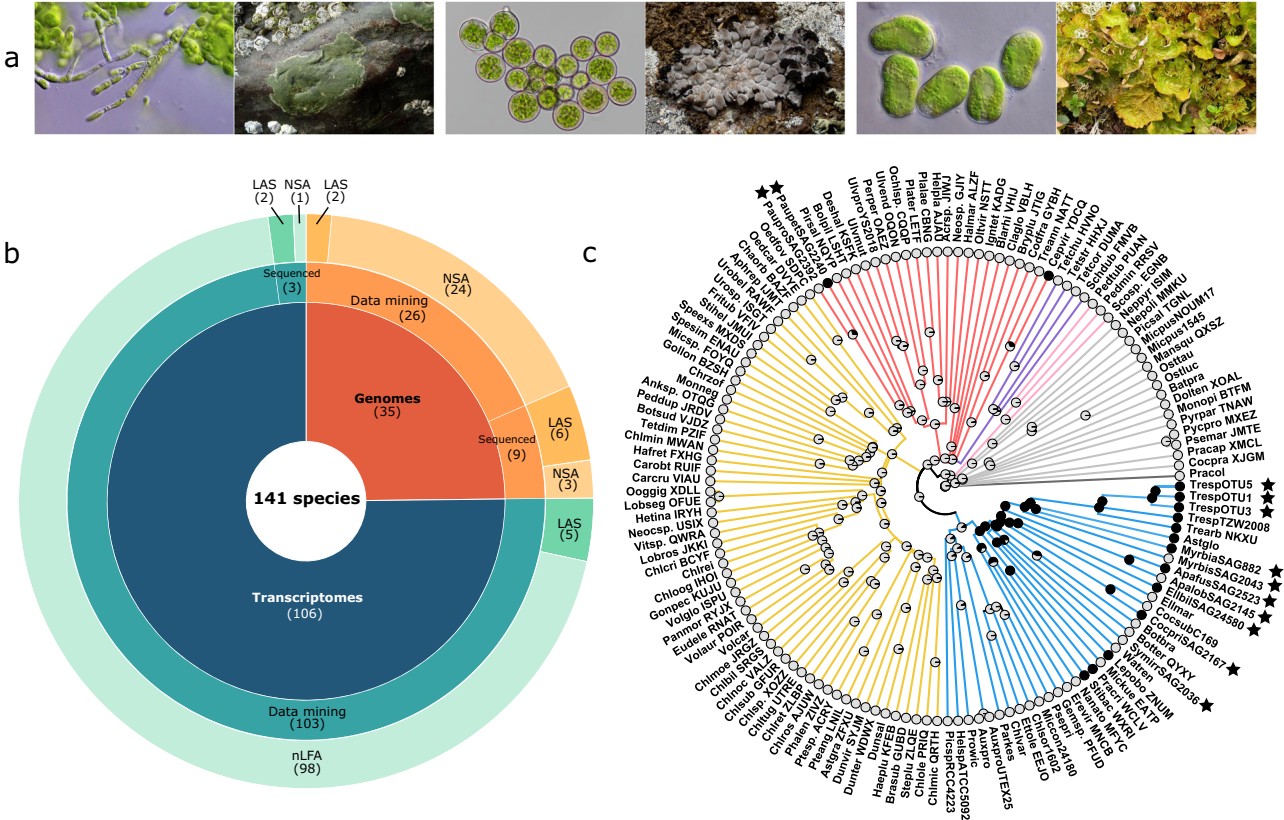

**Fig. 1 | Algal species genomes and transcriptomes sampling and ancestral state reconstruction. a** Pictures of three lichens and their algal partners: *Paulbroadya petersii and Verrucaria mucosa (right), Trebouxia sp* OTU1 and *Umbilicaria pustulata* (middle), *Myrmecia biatorellae and Lobaria linita* (right). **b** Algal genomes (oranges) and transcriptomes (blue) sampling, their sources, and their symbiotic habit (LAS lichen algal symbionts, NAS non-symbiotic algae). **c** Phylogenetic tree of Chlorophytes and ancestral state reconstruction of lichenization (ability to be involved in lichens) using the All Rates Different (ARD) model. LAS are indicated in black and NSA in gray. Black stars indicate the species that were sequenced for this study.

platform yielding an average of 40 million reads (details of sequencing and assembly statistics in Supplementary Data 3). The assembled transcriptomes reached 77.4% completeness when assessed by BUSCO.

The nine newly assembled genomes and three transcriptomes were combined with publicly available data mined from diverse databases, producing a final database of 141 species composed of 35 genomes (Supplementary Data 1) and 106 transcriptomes (Supplementary Data 1) of LAS and NSA species, covering all the chlorophyte classes.

**Ancestral state reconstruction supports at least three independent gains of the ability to associate with lichen fungal symbionts in Chlorophytes**

Textbooks and recent studies[20,23] all converge to a single hypothesis for the evolution of lichenization: it evolved in a convergent manner in Chlorophytes. Such evolutionary scenario had been proposed in the past for other type of symbioses, such as the nitrogen-fixing root nodule symbiosis[33], and later rejected[13,14,34]. The alternative hypothesis for the evolution of lichenization, a single gain followed by multiple losses, has so far not been explored. To determine the evolutionary history of lichenization in Chlorophytes, either rejecting the consensus convergent gains hypothesis or further supporting it, we conducted an Ancestral State Reconstruction (ASR) approach. The predicted proteomes from the 141 species were used as a matrix to reconstruct orthogroups using OrthoFinder, yielding a total of 2,157,361 genes (74.6% of the total) assigned to 197 669 hierarchical orthogroups (Supplementary Data 4). Based on the most informative orthogroups, a species tree of the 141 chlorophyte species was computed, rooted on *Prasinoderma coloniale*[35]. Either of two states for the lichenization

capacity (LAS or NSA) were assigned to each chlorophyte algae present in the sampling. The status of algal species as symbionts was assigned following[21]. Furthermore, given that many algal species lack distinct morphological features and their lichenization status has often only been reported in light microscopic studies, the determination of the lichenization status for each species in this study was based on a thorough review of published studies based on sequence data[36–41]. The ASR inferred three gains of lichenization within the Chlorophytes, one in Trebouxiophyceae and two in Ulvophyceae (Fig. 1). In Trebouxiophyceae, the single gain of the lichenization ability was followed by eleven putative losses in *Myrmecia bisecta* (SAG2043), *Apatococcus lobatus* (SAG2145), *Elliptochloris marina*, *Coccomyxa subellipsoidea* (C-169), *Botryococcus braunii*, *Botryococcus terriblis*, *Symbiochloris irregularis* (SAG2036), *Watanabea reniformis*, *Microthamnion kuetzingianum*, *Nannochloris atomus*, *Eremosphaera viridis* and *Geminella sp*. The more limited sampling in Ulvophyceae does not allow identifying loss events and, as well, may have masked additional gains. Based on the ASR, it can be proposed that the ability to engage in the symbiosis was acquired at least three times independently in Chlorophytes, aligning with the current consensus hypothesis for the evolution of this trait[23]. Although at the macroscopic level, the trait (*i.e.*, a chlorophyte alga hosted inside a fungal thallus) can be considered identical across the different clades, lichen symbiosis should be considered as a group of diverse symbioses rather than a single interaction.

**Trebouxiophyceae symbionts share conserved hierarchical orthogroups (HOG)**

Functional innovations are associated with the gains of genomic or genetic features which can be tracked by comparative genomics. Our

gathered dataset encompasses 13 LAS and 23 NSA in the Trebouxiophyceae class allowing to conduct such a comparative analysis to identify genes and gene families associated with the ability to engage into lichens. When comparing general genomic features such as protein coding gene number, GC and transposable elements contents, and genome size no differences were observed between symbiotic and non-symbiotic Trebouxiophyceae (Supplementary Fig. 1, Supplementary Fig. 2, Supplementary Data 5) apart from the GC content which is lower in symbiotic Trebouxiophyceae. Hence, the gain of the ability to lichenize did not involve massive genomic modifications as it is the case for other symbioses[29–32].

To identify genes potentially associated with lichenization in Trebouxiophyceae, we focused on the computed orthogroups for the entire Chlorophytes using two complementary statistical approaches. First, we conducted a sparse Partial Least Square Discriminant Analysis (sPLS-DA) in which each orthogroup composition is analyzed to identify orthogroups whose composition clusters the 141 species into two groups: the symbiotic Trebouxiophyceae and the other Chlorophytes. The first two principal components are responsible for 2 and 1% of the discrimination of the species into the two groups respectively (Fig. 2a). Since the first component is the most discriminant one, we focused on the 100 orthogroups that contribute the most to it (Fig. 2b, Supplementary Fig. 3). In a complementary approach, we applied a Mann–Whitney-Wilcoxon test to identify orthogroups that are significantly enriched in sequences from symbiotic Trebouxiophyceae. This approach identified 5 252 orthogroups (p value < 0.01, Fig. 2). When cross-referencing the sPLS-DA data with the Mann–Whitney-Wilcoxon test, we found a perfect overlap. Indeed, the 100 top orthogroups from the sPLS-DA were among the 5 252 orthogroups identified by the Mann–Whitney-Wilcoxon test (Fig. 2c, Supplementary Data 6). These 100 top orthogroups thus represent genes potentially associated with the evolution of lichenization in Trebouxiophyceae (Supplementary Data 7).

## Gene family expansions are associated with the evolution of lichenization in Trebouxiophyceae

To narrow down the most promising candidates associated with the origin of lichenization in Trebouxiophyceae and test their symbiotic relevance, the 100 candidate orthogroups were further analyzed. First, because genes involved in symbiotic association often show differential regulation in the presence of the other symbiont[2,42], we determined whether the expression level of the candidates was affected during lichenization. For this, we collected RNAseq data previously obtained for *Trebouxia sp*. TZW2008 grown in the absence of symbiotic fungus, in co-culture with the LFS *Usnea hakonensis*, or in well-established lichens[25] and recomputed differentially expressed genes. This analysis revealed a total of 3540 differentially regulated genes, either up- or down-regulated, when *Trebouxia sp*. TZW2008 associates with *Usnea hakonensis* (Supplementary Data 8)[25]. The differentially expressed genes were cross-referenced with the 100 orthogroups associated with lichenization in Trebouxiophyceae. Comparing the two datasets, we identified 42 orthogroups showing at least one *Trebouxia sp*. TZW2008 gene differentially regulated (14 HOG with up-regulated genes, seven with both up and down-regulated genes, and 21 with only down-regulated genes) in association with *Usnea hakonensis*.

Although OrthoFinder and other orthogroup-generating tools represent the only options to study genome-wide phylogenomic patterns, the resolution of the orthogroups is dependent on the sampled species and the gene-family complexity. In other words, orthogroups might either exclude actual orthologs or include non-orthologous genes. To reconstruct the evolutionary history of these candidate genes with higher confidence, we subjected them to phylogenetic analysis. Using targeted phylogenetic inference, five of the candidate genes were not found associated with lichenization anymore and five

of the phylogenies were not resolved enough to conclude on the evolutionary history of the genes (Supplementary Data 9).

Thus, from the 42 candidate orthogroups, a total of 32 showed phylogenetic and differential gene expression (in *Trebouxia sp*. TZW2008) patterns associated with the symbiotic habit. Reverse genetic analyses will be required in the future to validate their functions when a genetically tractable system and the in vitro resynthesis or lichen formation have been developed. Among the candidate genes associated with the symbiosis, eight contain genes that are annotated with IPR domains and can be associated with a putative function (Fig. 3, Supplementary Data 7). The 32 candidate orthogroups exhibit distinctive phylogenetic distributions, including indications of gene family expansions, exemplified by N0.HOG0002085, which encompasses genes annotated as glucose/ribitol dehydrogenase and short-chain dehydrogenase/reductase (SDR) (Fig. 3). Additionally, some candidates are LAS-specific, as seen in N0.HOG0012965, which contains a carbohydrate-active enzyme belonging to the glycoside hydrolase 8 family, or in N0.HOG0012501 that contains glutathione S-transferase enzymes (Supplementary Data 7). Furthermore, certain candidates such as N0.HOG0025580 and N0.HOG0025596 (both with an unknown function) display a specific distribution among Trebouxiales only (Fig. 3, Supplementary Data 9). Altogether, the phylogenomic comparison reveals that diverse genomic processes, including gene family expansions, contributed to the evolution of lichenization in the Trebouxiophyceae.

## Horizontal gene transfers contributed to the evolution of Trebouxiophycean lichens

Besides genes family expansion, two genes seemed to be highly specific to the symbiotic Trebouxiophyceae and almost completely absent from non-symbiotic Trebouxiophyceae and other Chlorophytes (Supplementary Data 9, Supplementary Fig. 4). Such evolutionary pattern can be the result of de novo gene birth or horizontal gene transfer (HGT). HGTs have been found previously as drivers for the acquisition of functional innovations across living organisms, including plants[43–45]. To determine the origin of these two symbiosis-associated genes, further phylogenetic analysis of the symbiotic Trebouxiophyceae-specific orthogroups was conducted, using additional databases including the main eukaryotic and prokaryotic lineages, to search for putative homologs across the tree of life. An origin by HGT was clearly identified for the two candidates. The first one, the orthogroup N0.HOG0012965, corresponds to an enzyme from the GH8 family. Based on the CAZy classification, GH8 enzymes have only been found in bacteria[46]. This orthogroup was ranked first in both the sPLSDA and the Mann-Whitney-Wilcoxon test (Fig. 2, Supplementary Data 6). Within the Trebouxiophyceae, GH8 are specifically present in LAS and in five NSA sister-species to well-characterized LAS (Supplementary Fig. 4, Supplementary Data 9). To ensure that the presence of the GH8 enzyme in LAS genomes was not due to potential contaminations, we scrutinized the scaffold they belong to. We found the GH8 well-anchored in their respective scaffolds, surrounded by algal genes. Re-mapping of the raw reads on these scaffolds excluded the possibility of chimeric scaffolds (Fig. 4b, Supplementary Fig. 5). The assignment of this orthogroup to the GH8 family was also confirmed by an unsupervised classification of carbohydrate-active enzymes using CUPP (Supplementary Data 10). The phylogenetic analysis identified GH8 members in bacteria, but also in 209 fungal species and strains (Fig. 4a, Supplementary Fig. 6, Supplementary Data 11), mostly from the non-symbiotic fungal phylum: the Mucoromycotina (Supplementary Data 11). Such a distribution could be explained by the presence of the GH8 enzyme clade in the eukaryotes most recent common ancestor, followed by losses and its specific retention in only two clades. However, such pattern would require losses in multiple eukaryotic lineages. The other hypothesis, the acquisition through an

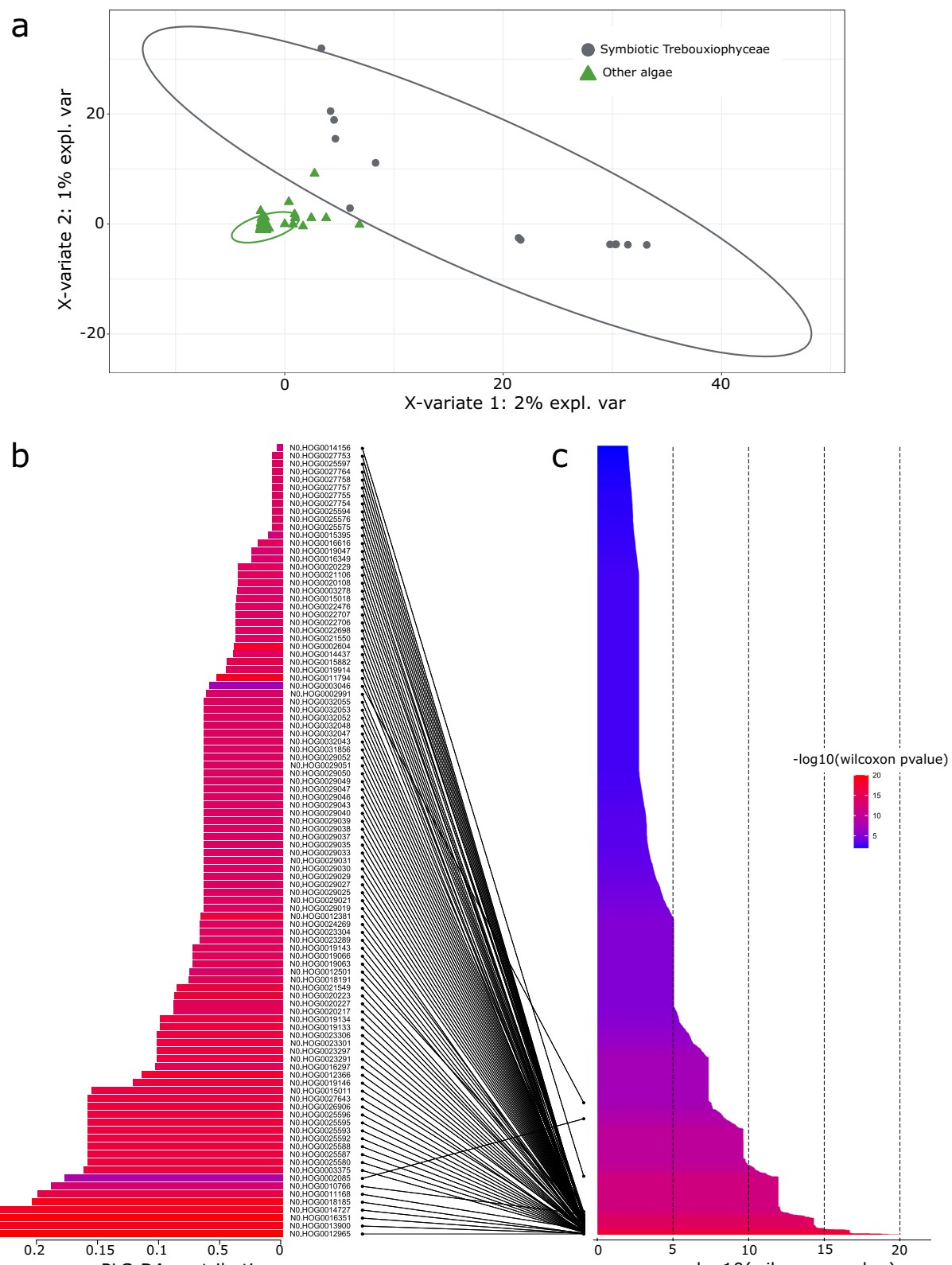

**Fig. 2 | Sparse PLS-DA and Mann–Whitney-Wilcoxon results. a** Individual sPLS-DA plot of the first two components discriminating chlorophyte species into two groups (gray: symbiotic Trebouxiophyceae, green: other algae) according to the 100 best orthogroups. **b** The 100 best orthogroups identified with the sPLS-DA and their contribution on the first component. **c** Barplots of the pvalues of the significant orthogroups using a two-sided Mann–Whitney-Wilxocon test (pvalue < 0.01 or −log10(pvalue) > 2). The bars are colored according to the Wilcoxon pvalues.

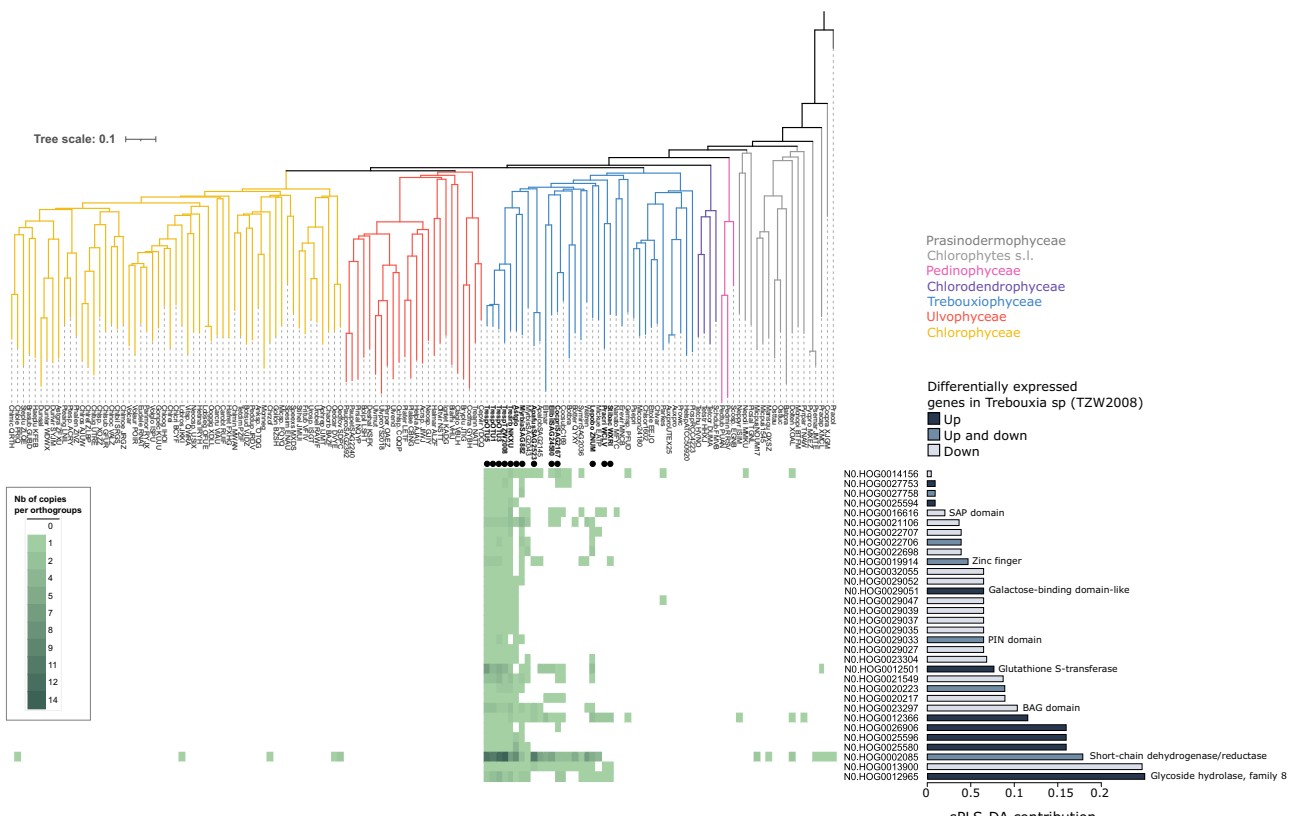

**Fig. 3 | Distribution of symbiotic-associated candidate genes in Chlorophytes.** Chlorophytes phylogeny, heatmap of the number of genes per species and per orthogroup for the ones that contain at least one differentially expressed gene (up and/or down-regulated) in symbiosis for *Trebouxia sp* (TZW2008) according to the data from Kono *et al.*, 2020, the contribution of each orthogroup according to

Fig. 2, the transcriptomic state of the differentially expressed genes in symbiosis and the main functional annotation of each orthogroup (all the IPR domains and GO terms found in the orthogroups are available in Supplementary Table 7). Symbiotic Trebouxiophyceae are indicated with black dots. The orthogroups without a functional annotation are listed as "unknown function".

HGT, is more parsimonious, requiring only two events. The phylogenetic analysis thus supports at least two HGT events in the evolutionary history of the GH8 family. The GH8 originated in bacteria and was horizontally transferred to fungi and Trebouxiophyceae independently, or, alternatively, the GH8 was first transferred to Mucoromycotina fungi as an intermediate recipient between bacteria and the algae (Fig. 4a, Supplementary Fig. 6). Following this HGT, the GH8 enzyme was retained in most symbiotic Trebouxiophyceae species (11/13) but lost in most species that did not maintain the ability to engage in the lichen symbiosis (5/23 non-symbiotic Trebouxiophyceae). According to the CAZy database, members of the GH8 family catalyze the hydrolysis of fungal polymers such as chitosan or lichenans found in LFS[47,48]. In bacteria, the GH8 family has been divided into three subfamilies based on the position of the proton acceptor residue[47,48]. Alignment of reference proteins from the three bacterial GH8 subfamilies with the GH8 sequences identified in symbiotic Trebouxiophyceae revealed that they share the asparagine catalytic site with the GH8b subfamily (Supplementary Fig. 5c) known to encompass, among others, lichenase enzymes able to degrade lichenan[48]. Additionally, the 3D model of the GH8 from symbiotic Trebouxiophyceae generated using AlphaFold2[49] positioned the catalytic asparagine in a pocket where the substrate could bind (Supplementary Fig. 7). To functionally test the enzymatic activity of the GH8 from Trebouxiophyceae, we cloned orthologs from three LAS and expressed them in a heterologous system. Among them, we successfully produced the recombinant GH8 from *Asterochloris glomerata*. Enzyme assays towards different polysaccharides confirmed a typical lichenase activity[50] for this enzyme with a significant cleavage of mixed linked β−1,3/β−1,4 glucans, while cleavage was neither observed on cellulose

nor on chitosan substrates (Fig. 4c). The detailed chromatographic analysis of the major soluble products that accumulated over time upon enzyme action showed they do not correspond to β−1,4-linked cellooligosaccharides nor to β−1,3-linked laminari-oligosaccharides, thus suggesting that the degradation products belong to the mixed linked β−1,3/1,4 class. These results suggest that the GH8 enzyme was acquired through horizontal gene transfer (HGT) in the MRCA of Trebouxiophyceae, along with the ability to interact with LFS. The enzyme was later retained in species that engage in symbiosis.

The second candidate, the N0.HOG0012501 orthogroup, consists of genes belonging to the glutathione S-transferase enzyme family. Phylogenetic analysis revealed that the algae sequences are nested within bacterial clades, indicating that the likely donors of the HGT are bacteria. Moreover, the original orthogroup is dispersed across two distinct bacterial clades, suggesting the possibility of a double transfer of a similar gene: one present in nearly all symbiotic Trebouxiophyceae (9/13) and only a few non-symbiotic Trebouxiophyceae (4/23), and another one that seems Trebouxia specific (Supplementary Fig. 8). Here again, the scaffold anchoring and the read mapping did not show any sign of contamination (Supplementary Fig. 9). This family is well known for containing enzymes involved in a wide range of biological processes[51] but more especially in buffering oxidative stresses.

## Discussion
Understanding the evolution of traits and the underlying genetic mechanisms has been studied in multiple contexts, from coat color in mice[52] to plant intracellular symbioses[22]. These genetic novelties may evolve through multiple mechanisms, from gene family expansion

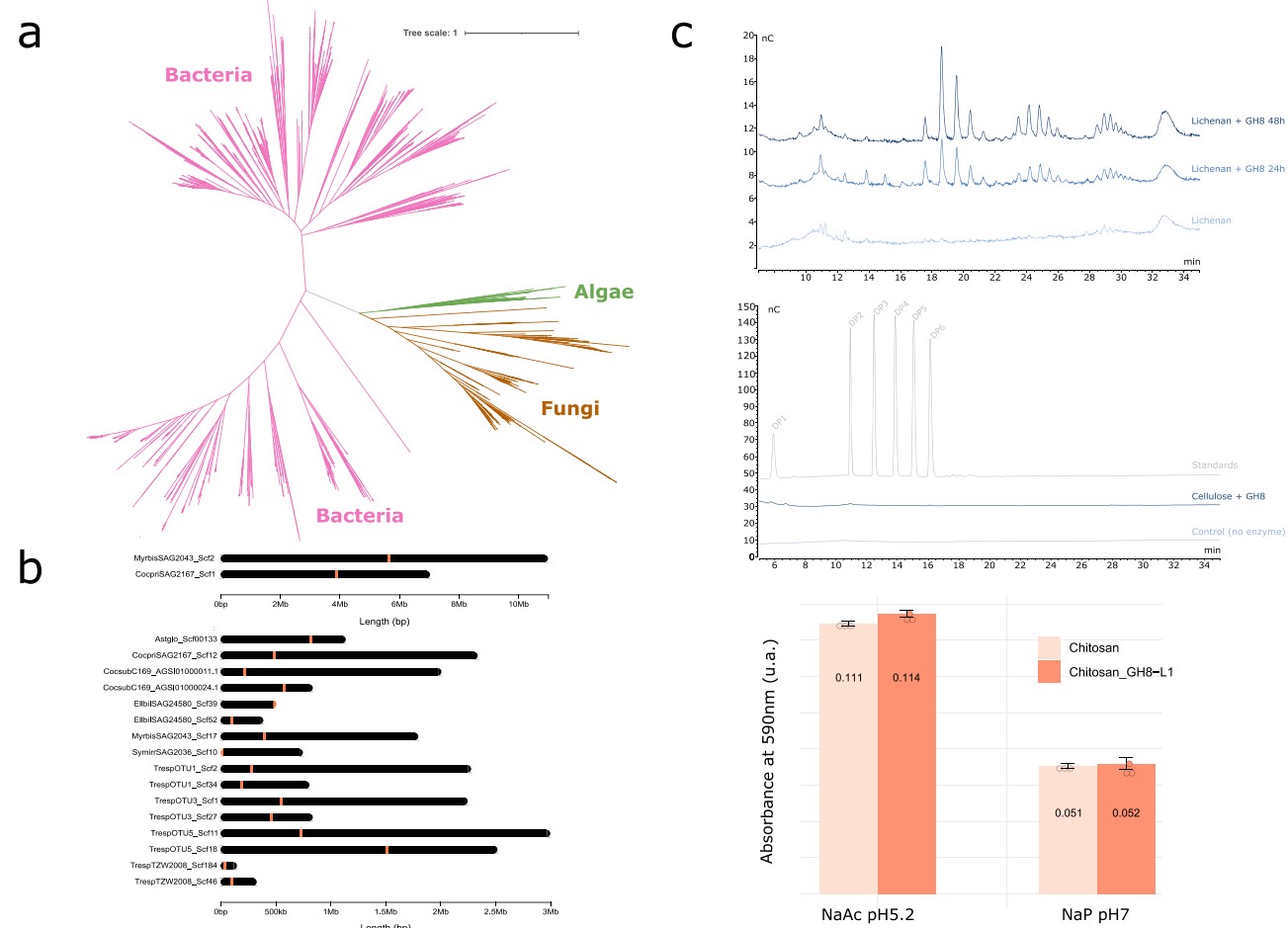

**Fig. 4 | Evolution of the GH8 enzyme in algae. a** Unrooted maximum likelihood tree of the sequences corresponding to the GH8 family. Branches are colored as follow: bacteria in pink, fungi in brown and chlorophyte algae in green. **b** GH8-like anchoring in scaffold of the different chlorophyte algae. The GH8 position is indicated by the orange lines. **c** Enzymatic activity of the GH8 enzyme from *A. glomerata* on, from top to bottom: lichenan, cellulose and chitosan (data presented as mean values ± standard deviation with *n* = 3 independant replicates).

mediated by duplications to de novo gene birth by domain fusion or horizontal gene transfer.

One of the candidate genes originating from gene family expansion is annotated as a short-chain dehydrogenase/reductase (SDR). SDR belongs to a family encompassing enzymes involved in ribitol biosynthesis[53]. Ribitol is an acyclic pentose alcohol previously identified as the major sugar produced in lichens such as *Peltigera aphthosa* (*Coccomyxa* photobiont), *Xanthoria aureola* or *Gyalolechia bracteate* formed with *Trebouxia* spp. photobionts[54,55]. The addition of exogenous ribitol to a culture of LFS has been shown to stimulate fungal growth and developmental transitions and has been suggested as a signal for lichen initiation, in addition to being a source of carbon for the LFS[54,56]. We speculate that expansion of the SDR gene family in Trebouxiophyceae may have enhanced ribitol biosynthesis, stimulating lichen morphogenesis and carbon transfer to the LFS.

Although considered rare in eukaryotic genomes, horizontal gene transfer is becoming a common theme in the evolution of innovations. In our study, we found that two genes associated with lichenization were horizontally acquired. The first one is annotated as a glutathione S-transferase. These enzymes are known to play a role in oxidative stress response in a wide range of organisms, including in lichen fungal and algal symbionts[57]. This class of enzymes was previously identified in lichen symbionts to play a role in desiccation resistance[57].

The other horizontally-acquired orthogroup is the one that best discriminates symbiotic Trebouxiophyceae from other Chlorophytes

at the phylogenetic level and belongs to the GH8 family. Contribution of HGT to the evolution of interactions between organisms has been reported in multiple eukaryotic taxa. Ferns and *Cycas* have gained the ability to produce toxins improving resistance to herbivores[58,59] and *Caenorhabditis elegans* detoxifies cyanogenic compound found in plants[60]. In addition, such HGT events have been also identified between the LFS *Xanthoria parietina* and its associated algal symbiont, *Trebouxia decolorans*. The latter has likely horizontally acquired three genes that could have played a role in the evolution of lichenization ability[61]. The xenologous origin of cell wall-degrading enzymes has been also observed in parasites, including phytophagous insects[62] and nematodes[63]. This preponderance of plant cell wall degrading enzymes HGT might reflect a more general mechanism for the evolution of inter-organism interactions. Diversification of the enzymes occurs in microorganisms evolving on a given substrate, such as tree bark, and horizontally transferred to other eukaryotes from the same ecological niches. This transfer expands the cell wall degradation potential of the recipient species and facilitate interactions, either mutualistic, endophytic or parasitic. In the classical model for the evolution of novelties, potentiation−actualization−refinement, potentiation can be considered here as a community phenomenon associated to the diversification of enzymes.

Independently of their origin, the role of cell wall degrading enzymes, and in particular Glycoside Hydrolases (GHs), in mutualistic interactions has been well documented. The current lack of genetic

model in LAS do not allow testing the biological role of the GH8 enzyme during lichenization, but a few hypotheses emerge based on the knowledge acquired on other symbiotic systems. First, the algal GH8 enzyme may play a crucial role in facilitating the establishment of a symbiotic interface between the fungus and the algae by breaking down the lichenan, a key component of the mycobiont's cell wall. Macrolichens consist of multiple layers, and lichenan has been identified predominantly in the medullary region of lichen species like *Cetraria islandica*[64]. In proximity to symbiotic Trebouxiophyceae, the fungal cell wall appears to be thinner[64], aligning with the ability of GH8 enzymes to break down lichenans present in this region. Other carbohydrate-active enzymes have previously been identified in the mycobiont *Usnea hakonensis*, and they are believed to be involved in creating a symbiotic interface by breaking down the algal cell walls (GH2, GH12), although these enzymes do not seem specific to LFS[25]. Overall, both the LFS and LAS appear to possess a genetic toolkit for degrading each other's cell walls[25,27]. Accommodation of micro-symbionts via modification of the host cell wall is well described in plant symbioses such as the arbuscular mycorrhizal and ectomycorrhizal symbioses[65], and for endophytism[17], a similar function can thus be proposed for lichens. A second potential role for the GH8 enzyme could be to generate a carbon source that could be utilized by other partners within the lichen thallus. This mechanism was previously identified in other symbiotic relationships, such as that observed in *Streblomastix strix*, an oxymonad residing in the termite gut alongside a bacterial community. In this context, bacteria employ their GHs to break down wood particles into monosaccharides, which they can subsequently assimilate[66]. Similarly, we can speculate that LAS engulfed in the macrolichens could generate simple sugars, in particular glucose monomers, from lichenans and use them as a carbon source as an alternative pathway to photosynthesis for carbon assimilation. The direct uptake of glucose by *Trebouxia* during symbiosis has been previously reported and proposed as an additional source of carbon to increase ribitol efflux[67]. Additionally, it was previously suggested that *Trebouxia* strains seem to adopt a heterotrophic lifestyle when cultured in the dark when a source of carbon such as glucose is provided in the culture medium. Hence, we can consider that LAS are mixotrophic and that the degradation of lichenans in simpler monomers could be used for their own nutrition[68]. As a working model, we propose that the combined gain of the GH8, increasing carbon availability to the LAS, and expansion of the SDR family, increasing ribitol biosynthesis, contributed to the evolution of lichenization in Trebouxiophyceae.

Our study expands the range of chlorophyte clades with sequenced genomes and revealed at least three independent origins of lichenization in green algae, confirming the convergent nature of this trait. We identified genes associated with lichenization originating from gene family expansion and horizontal gene transfers, including one enzyme able to degrade carbohydrate polymers formed in the thallus of lichen fungal symbionts. In the future, the development of genetic models in Trebouxiophyceae and lichen reconstitution in controlled conditions will allow proving the involvement of these genes in the symbiosis.

## Methods

### Algal cultures
Three *Trebouxia* photobionts with varied ecologies[69] were isolated from thalli of the lichen *Umbilicaria pustulata* using a micromanipulator as described in[70]. Cultures were grown on solid 3 N BBM + V medium (Bold's Basal Medium with vitamins and triple nitrate[71]) under a 30 µmol/m²/s photosynthetic photon flux density with a 12 h photoperiod at 16 °C. The identity of the isolated photobionts was validated by comparing the ITS sequence to those from[69]. Eleven additional algal cultures were obtained from the SAG Culture Collection (Göttingen). These represent a selection of both lichenized

algae as well as closely-related, free-living lineages—i.e. species that were never reported to establish symbiotic associations with fungi - belonging to the classes Trebouxiophyceae and Ulvophyceae (Chlorophyta) (Supplementary Data 1). The cultures were maintained under the conditions described above and sub-cultured every two to three months onto fresh medium until sufficient biomass (~500 mg) for DNA isolation was obtained.

### DNA isolation and sequencing
Prior to DNA isolation, we performed nuclei isolation to reduce the amount of organelle DNA, i.e. chloroplast and mitochondrial and non-target cytoplasmic components. This step has been shown to increase the read coverage of the targeted nuclear genomes and it is particularly recommended for long-read sequencing[72]. Green algae were transferred to fresh agar plates two days before nuclei isolation. For this, we used a modified protocol by Nishii et al. (2019, https://stories.rbge.org.uk/archives/30792) starting with 300–600 mg of algal material. Briefly, for each sample we prepared 20 ml of nuclei isolation buffer (NIB) consisting of 10 mM Tris-HCL pH 8.0, 30 mM EDTA pH 8.0, 100 mM KCl, 500 mM Sucrose, 5 mM Spermidine, 5 mM Spermine, 0.4% β-Mercaptoethanol, and 2% PVPP-30. The fine algal powder was transferred to 50 ml Falcon tubes with 10 ml ice-cold NIB and mixed gently. The homogenates were filtered into 50 ml centrifuge tubes through 20 µm cell strainers (pluriSelect, Leipzig, Germany), followed by a centrifugation at 2500 × g at 4 °C for 10 min. The pellets were resuspended in 9 or 9.5 ml NIB by gently tapping the tubes. 1 or 0.5 ml of 10% Triton X-100 diluted NIB (NIBT). After a 15 min incubation on ice, the suspensions were centrifuged at 2500 × g at 4 °C for 15 min. The nuclei pellets were carefully resuspended in 20 ml Sorbitol buffer (100 mMTris-HCL pH 8.0, 5 mM EDTA pH 8.0, 0.35 M Sorbitol, 2% PVPP-30, 2% β-Mercaptoethanol). After a 15 min centrifugation at 5 000 x g and 4 °C the supernatants were discarded, and the tubes were inverted on a paper towel to remove traces of buffer. After a RNAse A-/Proteinase K digestion for several hours the gDNAs were isolated following the protocol by[73] with modifications described in[74] or with Qiagen Genomic-Tips.

### Long-read DNA sequencing
SMRTcell libraries were constructed for samples passing quality control (Supplementary Data 2) according to the manufacturer's instructions of the SMRTcell Express Prep kit v2.0 following the Low DNA Input Protocol (Pacific Biosciences, Menlo Park, CA) as described in[74]. Genomic DNA was sheared to 20-kb fragments using Megaruptor 2 (Diagenode, Belgium) and then bead-size selected with AMPure PB beads (Pacific Biosciences) to remove <3-kb SMRTbell templates. SMRT sequencing was performed on the Sequel System II with Sequel II Sequencing kit 2.0 (Sequel Sequencing kit 2.1 for Sequel I system, see below) in 'circular consensus sequencing' (i.e., CCS) mode, 30 h movie time with pre-extension and Software SMRTLINK 8.0. Samples were barcoded using the Barcoded Overhang Adapters Kit-8A, multiplexed, and sequenced (3 samples/SMRT Cell) at the Genome Technology Center (RGTC) of the Radboud university medical center (Nijmegen, the Netherlands). Four samples were instead sequenced on the Sequel I system at BGI Genomics Co. Ltd. (Shenzhen, China) (Supplementary Data 2). In this case, one SMRT Cell was run for each sample.

### RNA isolation and sequencing
For RNA isolations we used the Quick-RNA Fungal/ Bacterial Miniprep Kit (Zymo Research) starting with 30-50 mg of algal material. RNAs were further purified, when necessary, with the RNA Clean & Concentrator-5 Kit (Zymo Research). Total RNAs from the 12 algal cultures (Supplementary Data 3) were sent to Novogene (Hong Kong, China) for library preparation and sequencing. mRNA-seq was performed on the Illumina NovaSeq platform (paired-end 150 bp sequencing read length).

## Genome assembly

Sequel II samples were demultiplexed using lima (v1.9.0, SMRTlink) and the options '--same --min-score 26 –peek-guess'. De novo assembly was carried out for each PacBio (Sequel/Sequel II) CLR subreads set using the genome assembler Flye (version 2.7-b1587)[75] in CLR mode and default parameters. Each assembly was polished once LAS part of the Flye workflow and a second time with the PacBio tool GCpp v2.0.0 with default parameters (v1.9.0, SMRTlink). The polished assemblies were scaffolded using SSPACE-LongRead v1.1[76] with default parameters.

The received scaffolds were taxonomically binned via BLASTx against the NCBI nr database (accessed in September 2020) using DIAMOND (--more-sensitive) in MEGAN v.6.7.7[77], with an e-value threshold of 1E-10 and the MEGAN-LR algorithm[77]. Only scaffolds assigned to the Chlorophyta were retained for subsequent analysis.

## Genome and transcriptome annotation

Genome assemblies were softmasked using Red[78] and annotated using BRAKER2 pipeline[79]. BRAKER2 was run with '--etpmode --softmasking --gff3 --cores 1' options. The pipeline in etpmode first train GeneMark-ETP with proteins of any evolutionary distance (i.e. OrthoDB) and RNA-Seq hints and subsequently trains AUGUSTUS based on GeneMark-ETP predictions. AUGUSTUS predictions are also performed with hints from both sources. The OrthoDB input proteins used by ProtHint is a combination of OrthoDB v10 (https://v100.orthodb.org/download/odb10_plants_fasta.tar.gz) and proteins from six species investigated in this study. To complement orthology-based annotation, available or generated RNA-Seq data for each species were used LAS hints in BRAKER2. Adapters and low-quality sequences were removed from the raw fastq files using cutadapt v2.1[80] and TrimGalore v0.6.5, (https://github.com/FelixKrueger/TrimGalore) with the options -q 30 --length 20. The cleaned reads were mapped against the corresponding genomes using HISAT2 v2.1.0[81] with the options --score-min L,−0.6,−0.6 --max-intronlen 10000 --dta. Duplicated reads were removed using the markdup command from SAMtools v1.10 [82]. These final alignments data were used LAS hints in BRAKER2[79].

We also annotated transcriptomes of four species (*Paulbroadya petersii, Paulbroadya prostrata, Myrmecia biatorellae, Stichococcus bacillaris*). First, we assembled the transcriptomes from the raw reads RNAseq using DRAP v1.92 pipeline[83]. runDrap was first used on the unique samples applying the Oases RNAseq assembly software[84]. Predictions of protein-coding genes were performed using TransDecoder v5.5.0[85] (https://github.com/TransDecoder/TransDecoder) and hits from BLASTp search in the Swissprot database (downloaded on September 2021) as well as HMMER search in the Pfam v34 database[86,87]. Completeness of newly sequenced and annotated genomes and transcriptomes was assessed using BUSCO V5.4.4[88] with default parameters and using the Chlorophyta "odb10" database (1,519 core genes) as reference. Transcriptomes from the 1KP project were also annotated following the same procedure with SwissProt downloaded on January 2019 and Pfam v32.

Finally, functional annotation was performed for all species investigated using the InterProScan suite v5.48-83.0[84] with the following analysis enabled: PFAM v33.1, ProSite profiles and patterns (2019_11), Panther v15.0, TIGERFAM v15.0, Gene3D v4.3.0, CDD v3.18, HAMAP 2020_05, PIRSF v3.10, PRINTS v42.0, SFLD v4.0, SMART v7.1 and SUPERFAMILY v1.75.

## Proteome database building

The corresponding proteomes of the 12 newly sequenced genomes and transcriptomes were added to a database that was built with proteomes extracted from public databases such as the NCBI and ORCAE (Supplementary Data 1). In total, the final database is composed of 141 species that cover all the chlorophyte clades and contains both LAS and NSA.

## Orthogroups reconstruction

Orthogroups reconstruction was performed using OrthoFinder v2.5.4[89] using the 141 species database with DIAMOND v0.9.19[90] set in ultra-sensitive mode. The estimated species tree based on orthogroups was then manually controlled and re-rooted on the outgroup species *Prasinoderma coloniale*. OrthoFinder was then re-run with this correctly rooted tree and with the MSA option to improve the orthogroups reconstruction (Supplementary Data 4).

## Ancestral state reconstruction

The ultrametric version of the 141 species tree obtained using Ortho-Finder was generated using the 'phytools' package[91] (v1.9.16) and used to perform an Ancestral State Reconstruction (ASR). All the species were coded as LAS or NSA. The ASR was then conducted using the 'ape'[92] package (v5.7-1) and plotted using the 'phytools' package (v1.9.16) in R (v4.2.2). Both equal rate (ER) and all rate different (ARD) models were tested, and the all-rate different model was retained based on the log-likelihood value.

## Genome streamlining investigation

Multiple symbionts exhibit strong genome reduction and modifications[29–32]. We compared genome size, the number of protein-coding genes, the GC content, and the transposable elements repertoire of LAS and NSA species. Each comparison has been performed using a Wilcoxon test using R v4.2.2[93].

## Transposable element annotation

We used EDTA v2.0.1[94] to annotate transposable elements. The EDTA pipeline combines an array of specific tools for different TE types, such as long terminal repeat transposons (LTRs): LTR_FINDER v1.07[95], LTR_retriever v2.6[96]; terminal inverted repeat transposons (TIRs): TIR-Learner v1.19[97]; Helitrons: HelitronScanner v1.1[98]; terminal direct repeats (TDRs), miniature inverted transposable elements (MITEs), TIRs, LTRs: Generic Repeat Finder v1.0[99]. It also runs RepeatModeler v2.0.3[100] to identify unknown TEs that were not found by the other tools. The pipeline then creates a filtered, combined repeat library of consensus sequences from the different sources, uses TEsorter[101] to de-duplicate and classify consensus sequences, and finally employs RepeatMasker v4.1.2-p1[102] to annotate the TEs in the genome. A widespread strategy is to use RepeatModeler alone to generate a library of consensus sequences and annotate them in the genome with Repeat-Masker. We opted to use EDTA instead for its multi-faceted approach and filtering procedure that produces a more informative repeat library, and thus a more detailed TE annotation in the genomes. EDTA was run with the options '--sensitive 1 --anno 1 --evaluate 1.

## Identification of genes associated with lichenization in Trebouxiophyceae

To identify genes linked to lichenization in Trebouxiophyceae, a sparse Partial Least Square Discriminant Analysis (sPLS-DA) was conducted using the 'mixOmics' package (v6.22.0)[103]. To do that, the 141 species were divided into two classes: the symbiotic Trebouxiophyceae and the other algae (including all the non-symbiotic species and the symbiotic Ulvophyceae). The orthogroup count (Supplementary Data 4) was then used as the quantitative dataset to identify the 100 orthogroups that discriminate the two classes best (Supplementary Fig. 3). To have a better resolution, the species-specific orthogroups and the orthogroups with two species were removed from the study. In parallel, a Wilcoxon test was conducted on the same orthogroup dataset to identify the ones that are enriched in symbiotic Trebouxiophyceae.

## Phylogenetic analysis of candidate proteins

To place expansions, contractions, and gene losses in an evolutionary context, candidate proteins were subjected to phylogenetic analysis.

First, homologs of sequences from orthogroups were searched against a database containing the 141 investigated species using the BLASTp v2.9.0 algorithm[104] and an e-value threshold of 1$^{e-10}$. Then, retained sequences were aligned using MUSCLE v5.1.0[105] with default parameters and obtained alignments cleaned with trimAl v1.4.1[106] to remove positions with more than 60% of gaps. Finally, alignments were used as a matrix for maximum likelihood analysis. First, phylogenetic reconstructions have been performed using IQ-Tree v2.1.3[107] with default parameters to obtain a global topology of the tree. Before tree reconstruction, the best-fitting evolutionary model was tested using ModelFinder[108] implemented in IQ-TREE. Branch supports were tested using 10 000 replicates of both SH-aLRT[109] and ultrafast bootstrap[110]. Trees were visualized and annotated in the iTOL v6 platform[111]. All candidate trees are provided in Supplementary Data 12.

### Identification of genes differentially expressed genes between the symbiotic state of Trebouxia sp TZW2008

The raw reads were downloaded and submitted to the nf-core/rnaseq v3.4[112] workflow in nextflow v21.04[113] using '-profile debug,genotoul --skip_qc --aligner star_salmon' options. Nextflow nf-core rnaseq workflow used bedtools v2.30.0[114], cutadapt v3.4[80] implemented in TrimGalore! v 0.6.7, picard v2.25.7 (https://broadinstitute.github.io/picard), salmon v1.5.2[115], samtools v1.13[116], star v2.6.1d and v2.7.6a[117], stringtie v2.1.7[118] and UCSC tools v377[119].

The counted data were analysed using *edgeR* package v2.1.7[120] with R v4.1.1[93]. Two samples of synthetic lichen showed distant clustering to other synthetic lichen samples (DRR200314 and DRR200315, named Tresp_LicSynt_R1 and Tresp_LicSynt_R2 respectively), so we decided to remove them. Then, we removed consistently low expressed genes with less than 10 reads across each class of samples (Algal culture, Synthetic lichen, and Field lichen). After, gene counts were normalized by library size and trimmed mean of M-values (i.e. TMM) normalization method[121]. We estimated differentially expressed genes (DEGs) by comparing synthetic lichen samples and field lichen samples to algal culture. DEGs were considered with adjusted p-value (FDR method) <0.05 and |logFC| >1.5 (Supplementary Data 8).

### Horizontal Gene Transfer demonstration

Three different approaches were used to validate the putative horizontal gene transfer of the GH8. First, the GH8-coding gene of algae was verified to be anchored in large scaffolds and surrounded by other algal genes. Visualization of GH8-like positions on scaffolds was performed using the R package chromoMap v0.3.1[122]. Secondly, reads from sequencing were mapped on the region containing the algal GH8 enzyme to control for chimeric assembly using minimap2 v2.17-r941[123] and default parameters. Finally, a phylogenetic analysis was conducted to place algal GH8 in an evolutionary context. Using the BLASTp v2.13.0.1+ algorithm[104] with an e-value threshold of 1$^{e-30}$ homologs of algal GH8-like were searched for in three different databases: the JGI fungal resources (accessed in February 2020, contains more than 1600 fungal genomes) and the non-redundant protein database from NCBI (May 2022) and the algae transcriptomes from the one KP project. Obtained sequences were subjected to phylogenetic analysis as described above and using MUSCLES[105] (v5.1.0) Super5 option for the alignment step and FastTree v2.1.10[124]. The presence of the GH8 functional domain was determined using hmmsearch from the HMMER v3.3.1 package[125] with default parameters and using the GH8 domain model (Pfam accession: PF01270). Protein structure was predicted using AlphaFold v2.1.0[49].

### Identification of carbohydrates active enzyme using CUPP

To identify all the carbohydrate active enzymes in Chlorophytes, CUPP v4.0.0[126] was used with default parameters and with the 2023 CUPP library.

## GH8 enzyme activity analysis

**Recombinant production of GH8 enzyme.** From all the LAS GH8, we selected the protein from *A. glomerata* to test its enzymatic activity. Its nucleotide sequence was first codon optimized for Pichia pastoris. The gene was synthesized by Genewiz (South Plainfield, New-Jersey, USA) and inserted in the expression vector pPICZαA (Invitrogen, Carlsbad, California, USA) in frame with the C-terminal poly-histidine tag. Transformation of competent *P. pastoris* X33 cells (Invitrogen, Carlsbad, California, USA) was performed by electroporation using the PmeI-linearized pPICZαA recombinant plasmid as described in[127] using the facilities of the 3PE Platform (Pichia Pastoris Protein Express; www.platform3pe.com/). Zeocin-resistant transformants were then screened for protein production. The best-producing transformant was grown in 4 L BMGY medium (10 g.L-1 glycerol, 10 g.L-1 yeast extract, 20 g.L-1 peptone, 3.4 g.L-1 YNB, 10 g.L-1 ammonium suASte, 100 mM phosphate buffer pH 6 and 0.2 g.L − 1 of biotin) at 30 °C and 200 rpm to an optical density at 600 nm of 2–6. Expression was induced by transferring cells into 800 mL of BMMY media at 20 °C in an orbital shaker (200 rpm) for another 3 days. Each day, the medium was supplemented with 3% (v/v) methanol. The cells were harvested by centrifugation, and just before purification, the pH was adjusted to 7.8 and was filtrated on 0.45-μm membrane (Millipore, Burlington, Massachusetts, USA).

**Purification by affinity chromatography.** Filtered culture supernatant was loaded onto a 20 mL HisPrep FF 16/10 column (Cytiva, Vélizy-Villacoublay, France) equilibrated with buffer A (Tris-HCl 50 mM pH 7.8, NaCl 150 mM, imidazole 10 mM) that was connected to an Äkta pure (Cytiva). The (His)6-tagged recombinant protein was eluted with buffer B (Tris-HCl 50 mM pH 7.8, NaCl 150 mM, imidazole 500 mM). Fractions containing the recombinant protein were pooled, concentrated, and dialyzed against sodium acetate buffer 50 mM, pH 5.2. The concentration of the purified protein was determined by absorption at 280 nm using a Nanodrop ND-200 spectrophotometer (Thermo Fisher Scientific) with calculated molecular mass and molar extinction coefficients derived from the sequence.

**Substrate cleavage assays.** The enzymatic activity of *A. glomerata* GH8 was tested on different types of substrates, i.e. cellulose (Avicel), lichenan (β−1,3/1,4-glucan), and chitosan. All substrates except Avicel cellulose (PH-101, Sigma-Aldrich), were purchased from Megazyme (Bray, Ireland). Amorphous cellulose (Phosphoris acid swollen cellulose or PASC) was prepared from Avicel cellulose as described in[128]. Enzyme assays were performed in a total volume of 200 μL containing 1% (w/v) polysaccharides or 0.5 mM of oligosaccharides in 50 mM pH 7.0 sodium phosphate buffer with 4 μM of *A. glomerata* GH8. The samples were incubated in a thermomixer (Eppendorf) at 30 °C and 1000 rpm, for 24-48 h. The samples were then boiled for 10 min to stop the enzymatic reaction and centrifuged at 15,000 × *g* for 5 min. The enzyme reactions were analyzed by high-performance anion-exchange chromatography coupled with pulsed amperometric detection (HPAEC-PAD) (Dionex ICS6000 system, Thermo Fisher Scientific, Waltham, MA, USA). The system is equipped with a CarboPac-PA1 guard column (2 × 50 mm) and a CarboPac-PA1 column (2 ×250 mm) kept at 30 °C. Elution was carried out at a flow rate of 0.25 mL.min-1 and 25 μL was injected. The solvents used were NaOH 100 mM (eluent A) and NaOAc (1 M) in 100 mM NaOH (eluent B). The initial conditions were set to 100% eluent A, following gradient was applied: 0−10 min, 0−10% B; 10−35 min, 10−30% B; 35−40 min, 30-100% B (curve 6); 40−41 min, 100−0% B; 41−50 min, 100% A.

## Reporting summary

Further information on research design is available in the Nature Portfolio Reporting Summary linked to this article.

## Data availability

Genome and transcriptome data from this study were deposited in NCBI under the BioProject PRJNA790449. The following database were used in this study: NCBI NR (accessed September 2020 for genome annotation and May 2022 for HGT investigation), Pfam v32 (1KP transcriptome annotation) and v34 (this study transcriptome annotation), OrthoDB v10, Mycocosm (last accessed in February 2020) and SwissProt (last accessed September 2021 for transcriptome annotation from this study and January 2019 for transcriptome annotation from the 1KP project). Publicly available genomes used in this study can be found in the NCBI under the following accession codes: *Auxenochlorella prototothecoides* 710 [GCF_000733215.1], *Auxenochlorella prototothecoides* UTEX25 [GCA_003709365.1], *Chlorella sorokiniana* [GCA_002245835.2], *Chlorella variabilis* NC64A [GCF_000147415.1], *Helicosporidium sp.* ATCC 50920 [GCA_000690575.1], *Micractinium conductrix* SAG241.80 [GCA_002245815.2], *Parachlorella kessleri* iCA-BeR21 [GCA_015712045.1], *Prototheca wickerhamii* HMC1 [GCA_003255715.1], *Ulva prolifera* [GCA_004138255.1]. *Picochlorum sp.* RCC4223 and *Ulva mutabilis* genomes were retrieved from ORCAE database respectively at https://bioinformatics.psb.ugent.be/gdb/RCC4223/ and https://bioinformatics.psb.ugent.be/gdb/ulva/. Transcriptomes of *Pseudochlorella pringsheimii, Watanabea reniformis, Elliptochloris marina* were assembled from SRA available in the NCBI under the respective series of SRA accession codes: [SRR11611235, SRR11611236, SRR11611237, SRR11611238], [SRR16849198] and [SRR3952294, SRR5133332]. Annotations of the 1KP transcriptomes have been deposited in FigShare under the DOI: 10.6084/m9.figshare.25611138.

## Code availability

No custom code has been produced in this study.

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

## Acknowledgements

We thank the genotoul bioinformatics platform Toulouse Occitanie (Bioinfo Genotoul, https://doi.org/10.15454/1.5572369328961167E12) for providing computing resources. J.K., C.L. and P-M.D are supported by the project Engineering Nitrogen Symbiosis for Africa (ENSA) currently funded through a grant to the University of Cambridge by the Bill & Melinda Gates Foundation (OPP1172165) and the UK Foreign, Commonwealth and Development Office as Engineering Nitrogen Symbiosis for Africa (OPP1172165). This project has received funding from the European Research Council (ERC) under the European Union's Horizon 2020 research and innovation program (grant agreement No 101001675

- ORIGINS) to P-M.D. This work was supported by the "Laboratoires d'Excellence (LABEX)" TULIP (ANR-10-LABX-41)" and by the "École Universitaire de Recherche (EUR)" TULIP-GS (ANR-18-EURE-0019). F.D.G. was supported by the LOEWE-Center for Translational Biodiversity Genomics (TBG) funded by the Hessen State Ministry of Higher Education, Research and the Arts (HMWK). We thank Carola Greve for assistance with PacBio library preparation, Anjuli Calchera for bioinformatic support, Andreas Beck for assistance with the micromanipulator algal cell isolation, Nicolas Piganeau and Bas Tolhuis for their helpful advice on PacBio sequencing and technical support.

## Author contributions

FDG, PMD and JK designed the project. JK, FDG, CP, CL, PS, MP, MH, SG and JO conducted experiments. JK, FDG, PS, CP, CL, JGB and PMD analyzed data. CP, JK and PMD wrote the manuscript with inputs from all authors.

## Competing interests

The authors declare no competing interests.
