## [Peer Review File · Nature Communications]

Phylogenomics reveals the evolutionary origins of lichenization in chlorophyte algaeREVIEWER COMMENTS

Reviewer #1 (Remarks to the Author):

Review of Puginier et al.

This is an important comparative genomics study looking at the origin of lichen symbiosis from the standpoint of symbiotic algae in the Trebouxiophyceae. This paper is built around three analyses and reported results: a phylogenomic reconstruction of algae involved in lichens and algae outside of lichens, with an ancestral state reconstruction; a comparative genomics analysis of lichen algae with those outside of lichens; and the study of a carbohydrate-active enzyme of the GH8 family described as a "lichenization gene".

I have seen a version of this paper previously on BioRxiv and welcome the application of comparative genomics and orthogroup analysis in aeroterrestrial algae in and out of symbiosis. Clearly some of the analyses are novel and likewise key elements of the results are important and need to be out there and published so they can be further tested. However, I have questions about whether some of the results are novel, and/or I am missing in-depth analysis and exploration of alternative hypotheses for some of the observed patterns. Specifically:

1) The authors report the independent origin of lichen symbiosis in algae, rejecting the hypothesis that lichen algae descended from a single origin. However, I am not aware anybody has ever proposed that lichen algae come from a single origin, and the result that lichen symbiosis arose more than once even in the algal tree is consistent with what has been stated e.g. in recent reviews of the topic (detailed comments below). The tree and ASR are novel, but it does not resolve an open question. This analysis feels like an odd add-on to this paper; the *Paulbroadya* and *Cephaleuros* are not incorporated into the other analyses and not really followed up on in any other way.

2) The orthogroup analysis identifies a large number of genes but only one is really discussed, and is somewhat oddly referred to as a GH8 gene (GHs are CAZy families that include genes coding for a variety of enzymes that include e.g. endoglucanases, so I guess it could be a gene coding for an endoglucanase classified as a GH8?). The alphafold and functional characterization of the enzyme is important but it is quite a stretch to call this and all the others lichenization genes (line 181). All we know is that elevated levels of mRNA are found in lichen algae and/or expressed in the lichen state, but the latter is also true of e.g. KP4 toxins in other lichen symbiosis studies; does that make them also lichenization genes?

3) The authors state that the gene they refer to as a GH8 is a new find in eukaryotes and "clearly" acquired from bacteria, but the tree of GH8 sequences plus their candidates from eukaryotes show the eukaryotic sequences as sibling to all other GH8 sequences, not derived. How did they arrive at the GH8 annotation? Also, the branch length between the algal and related and newly "discovered" fungal sequences are so short that I have more questions about the relationship of the putative algal GH8 to those in fungi than to anything in prokaryotes. To what extent have the authors explored alternative hypotheses here?

4) On a related note, if I am reading your results correctly you report cleavage of both beta-1,3 and beta-1,4 linkages but GH8s cleave only beta 1,4 linkages (including beta-1,4 linkages in mixed linkage molecules). Have you explored a relationship to other CAZy families involving mixed beta 1,3/1,4 linkages, such as GH131 (in which both are cleaved, and which occur in fungi)? How sure are you this is a GH8?

5) The orthogroup analysis identified many gene models but almost all the attention is focused on one gene coding for a putative GH8 endoglucanase. A glutathione S transferase is briefly mentioned, which is interesting in light of the key and known role glutathione regulation plays in lichen symbiosis (PNAS 102: 3141–3146). In general, I would expect the orthogroups to be much more thoroughly explored and discussed, especially in also in the context of at least some known pathways and signaling cascades. What % of the orthogroup gene models had annotations?

6) The paper is logically and sequentially structured, but I was surprised how vague the language is in key places, repeatedly stating that it will address our "elusive understanding" of lichen algae, but not clearly outlining the study objectives, questions, or hypotheses around the study system. I am missing crisply formulated language addressing known, published hypotheses around lichen algae and a review of their postulated role in and benefit from lichen symbiosis, and gaps the authors hope to fill with the present study. The introduction and especially the first paragraph of the discussion reach quite far into other systems for analogs, but the substantial literature on trebouxiophyte algae (1950s to the present) is barely discussed. For instance, the authors briefly speculate that a GH8 enzyme could be useful in acquiring carbon from sources other than photosynthesis, but do not review the literature in which trebouxioid mixotrophy has been observed and/or it has been speculated (e.g. by Ahmadjian 2002, book chapter in Seckbach, ed. Symbiosis: mechanisms and model systems. Cellular origin and life in extreme habitats) that trebouxioid algae derive carbon heterotrophically from fungi. This would seem to be quite relevant to their findings.

7) I found the frequency with which imprecise language is used adds up to an impediment to reading the paper. The authors talk about "transcriptomic state" instead of differential expression, and "lichen-forming", etc. when this study does not address morphogenesis and there is no new or existing evidence about any role the microbes play in lichen formation. Other examples in the specific comments.

In summary, I think there are solid analyses at the core of this study with some genuinely novel and exciting results but I feel like the authors do not sufficiently relate these to existing questions on lichen symbiosis and focus on one gene while only shallowly exploring the rest of their data. I especially hope that the authors can show that they have explored a wider range of possibilities regarding the origin and even annotation of the putative GH8.

Specific comments:

Abstract: first opening sentences ("our understanding of their evolution and function remains elusive") are way too vague and do not really set up the question behind the study

Independent gains of symbiosis in Trebouxiophyceae and Ulvophyceae really new here?

Abstract, line 38: GH8 is a family of GHs and not one specific enzyme, as implied here

Line 52: either

Line 52: why lichen-forming algae? There is no evidence that the algae form lichens, so perhaps something simpler, like lichen algae, or algal photobionts, would be appropriate

Line 61: what does "successfully unraveled the evolution" mean?

Line 65: why lichen-forming fungi? Again, neither partner is known to be able to "form" lichens, and this paper is not about that, so "lichen fungi" or "lichen fungal symbionts" might be more appropriate

Line 66: lichen studies

Line 67: what was studied is fungal involvement in lichens, not the ability to "form" them, which has not been studied (morphogenesis)

Line 71-72: again, this study is not about lichen-*** forming *** ability, which implies morphogenesis. It simply addresses occurrence in lichens

Line 71-73: who has ever proposed that the common ancestor of Trebouxiophyceae and Ulvophyceae was involved in lichens? I am not aware anyone has, and none of the citations provided (12,13,22,2) support this statement. It would be better to acknowledge that this is being

proposed as a hypothesis de novo in this manuscript, but I have my doubts whether there are really reasonable grounds to hypothesize this.

Line 77: the English is not clear ("constrained so far analyses")

Line 79: "the evolutionary history of lichens on the green algal side and the underlying molecular mechanisms remains elusive" — is the objective of this paper to redress this problem? The objectives and hypotheses should be clearly stated

Line 81: I can deduce what nLFA stands for but it is never explained

Line 82: not clear what is meant by "129 previously reported genomes (n = 26)"; I see that the 129 is the sum of 26+103 but perhaps this could be worded as "129 genomic datasets / species comprised of 26 genomes and 103 transcriptomes". Also, these numbers do not agree with those given on lines 108-109 (35+106 = 141).

Line 119: surely some steps are being left out here, as OrthoFinder does not generate phylogenetic trees

Line 121: what does it mean that "symbiotic abilities were defined based on 21 and finally curated by the authors of this paper." Perhaps better "the status of algal species as symbionts was assigned following Sanders & Matsumoto21" — please explain what the additional curation means

Lines 131-136: It has previously been observed that lichen symbioses are distinct phenomena polyphyletic in both the fungal and algal phylogenies (e.g. New Phytologist 234: 1566-1582). However, I fail to see how the evidence acquired in the present ASR supports the interpretation that "molecular mechanisms ... likely differ". The authors did not study this.

Lines 139-140: first sentence can be deleted

Line 161-163: exactly, this should be clearly stated as a caveat throughout

Line 164: "associated" is better than "linked"

Line 172: you state that you "recomputed" the DE genes, but in Table S8 the data are indicated as being from Kono et al.. Is this a new analysis?

Line 194: please replace "glycosyl hydrolase of the eighth family" with a more standard explanation that GHs are classified in the CAZy database as families and this is GH family 8. Also please get someone to revise this manuscript who speaks English fluently

Line 253-254: this doesn't belong in the results

Line 261: please don't call GH8 a gene, it is a CAZy family

Line 263: cycads

Line 268: bark (can't be made plural)

L256-274: it is good to place these findings in context but this paragraph becomes rambling and goes off topic, addressing issues such as potentiation etc. that have no relevance to what has been shown with this study

L288: provide a citation for the fungus possessing enzymes to break down algal cell walls

L296ff.: "we can speculate that LFA engulfed in the macrolichens could generate simple sugars from lichenans and use them as a carbon source as an alternative pathway to photosynthesis for carbon assimilation" — there is a fair amount of experimental literature on Trebouxia using external carbon sources, this would be good to tie into here with some citations

L371: please disclose how many of the cultures were axenic versus containing some bacteria or fungi

L468: what does "crossing" mean here?

L504: "the protein A. glomerata" — something is wrong or missing here, A. glomerata is one of your species

L563: authors?

Page 29: Figure 2A is unreadable

Page 30: I am confused by the plotting of orthogroups and results of differential expression analysis (called here "transcriptomic state"). If I understand correctly, the DE analysis is in fact a reanalysis of the Kono et al. data for just one of the black-dotted Trebouxia strains, right? Am I to understand that each orthogroup numbered n0.hog... contains/includes at least one gene model with log fold change differences (?) indicated on the unlabeled scale bottom right? I am especially confused how you juxtapose differential expression statistics which presumably refer to a single gene model in one strain next to mapped orthogroups which contain many genes. This is at a minimum very confusing and unreadable in its present form, and if it is plotted the way I think it is it is also misleading.

Page 31: I am not seeing the black dots or green line the authors refer to

Reviewer #2 (Remarks to the Author):

Here, the authors provide novel and critical insight into lichen evolution from the perspective of the photobiont. Their findings of the combined roles gene family expansion and horizontal gene transfer are particularly insightful. This work provides valuable, original insight into evolutionary genomics in one of the most iconic symbiotic systems. Data collection, analyses and interpretations are appropriate; and the claims and conclusions are supported by the data. Any the methodology is described in enough detail for work to be reproduced.

I recommend the study for publication.

RESPONSE TO REVIEWERS' COMMENTS

[NOTE: the line numbers correspond to the manuscript with track changes]

Reviewer #1 (Remarks to the Author):

Review of Puginier et al.

This is an important comparative genomics study looking at the origin of lichen symbiosis from the standpoint of symbiotic algae in the Trebouxiophyceae. This paper is built around three analyses and reported results: a phylogenomic reconstruction of algae involved in lichens and algae outside of lichens, with an ancestral state reconstruction; a comparative genomics analysis of lichen algae with those outside of lichens; and the study of a carbohydrate-active enzyme of the GH8 family described as a “lichenization gene”.

I have seen a version of this paper previously on BioRxiv and welcome the application of comparative genomics and orthogroup analysis in aeroterrestrial algae in and out of symbiosis. Clearly some of the analyses are novel and likewise key elements of the results are important and need to be out there and published so they can be further tested.

We thank the reviewer for this positive feedback on our work!

However, I have questions about whether some of the results are novel, and/or I am missing in-depth analysis and exploration of alternative hypotheses for some of the observed patterns. Specifically:

1) The authors report the independent origin of lichen symbiosis in algae, rejecting the hypothesis that lichen algae descended from a single origin. However, I am not aware anybody has ever proposed that lichen algae come from a single origin, and the result that lichen symbiosis arose more than once even in the algal tree is consistent with what has been stated e.g. in recent reviews of the topic (detailed comments below). The tree and ASR are novel, but it does not resolve an open question. This analysis feels like an odd add-on to this paper;

The convergent evolution hypothesis is indeed the preferred, and only formulated, hypothesis to explain the distribution of the ability to engage in lichens in Chlorophytes. However, theoretically, the single origin + multiple losses hypothesis can be proposed too. A similar situation was faced by the community working on the nitrogen-fixing root nodule symbiosis a few years ago: the convergent gain hypotheses was the consensus, but two phylogenomic studies rejected it.

It was essential for our work to exclude this alternative scenario as it would have impaired all the subsequent comparative phylogenomic approaches to identify the genetic features associated with the gain of the ability to lichenize.

We rephrased the section (lines 136-143) to reflect that i) the current consensus is clearly that lichenization evolved in a convergent manner in Chlorophytes and ii) we are testing whether an alternative hypothesis could be considered.

As the ASR strongly supported the textbook hypothesis, we did not include an additional experiment we have conducted to test for the “single gain + multiple losses hypothesis”. As for other symbioses (Delaux et al. 2014, Griesmann et al. 2018) we speculated that, if the “single gain + multiple losses hypothesis” had been correct we would have detected pattern of convergent gene losses, following

the co-elimination principle. We did not detect such convergent losses, further excluding our alternative hypotheses.

The Paulbroadya and Cephaleuros are not incorporated into the other analyses and not really followed up on in any other way.

The quality of these datasets is not suitable for the comparative phylogenomics.

2) The orthogroup analysis identifies a large number of genes but only one is really discussed, and is somewhat oddly referred to as a GH8 gene (GHs are CAZy families that include genes coding for a variety of enzymes that include e.g. endoglucanases, so I guess it could be a gene coding for an endoglucanase classified as a GH8?).

Please find our detailed reply to the questions relative to the GH8 identification/annotation and characterization below in response to comment #4.

The alphafold and functional characterization of the enzyme is important but it is quite a stretch to call this and all the others lichenization genes (line 181). All we know is that elevated levels of mRNA are found in lichen algae and/or expressed in the lichen state, but the latter is also true of e.g. KP4 toxins in other lichen symbiosis studies; does that make them also lichenization genes?

To define genes as “lichenization genes” we used three criteria:

- i) The orthogroup should be statistically associated with lichenization in the genome-wide comparative phylogenomic analysis (both PLSDA and Mann-Whitney-Wilcoxon tests). In other words: they discriminate LFA and nLFA. This is not the case for the KP4 toxin for instance.
- ii) The gene should have a phylogenetic pattern validated by targeted phylogeny (not only orthofinder-based)
- iii) The ortholog(s) of *Trebouxia* sp. TZW2008 should be differentially regulated in the interaction with *Usnea hakonensis*.

We acknowledge the fact that the actual symbiotic role of these genes during lichenization will have to be determined, and we state it in the revised version of the manuscript (lines 227-231):

“Thus, from the 42 candidate orthogroups, a total of 32 showed phylogenetic and differential gene expression (in *Trebouxia* sp. TZW2008) patterns associated with lichenization. Although reverse genetic analyses will be required in the future to validate their functions, when a genetically tractable system and the in vitro resynthesis or lichen formation will have been developed, we define these candidates as lichenization genes.”

Similar description as “symbiotic genes” have been made for genes in other contexts (Arbuscular mycorrhizal symbiosis: Bravo et al. 2016; root nodule symbiosis: Libourel et al. 2023).

3) The authors state that the gene they refer to as a GH8 is a new find in eukaryotes and “clearly” acquired from bacteria, but the tree of GH8 sequences plus their candidates from eukaryotes show the eukaryotic sequences as sibling to all other GH8 sequences, not derived. How did they arrive at the GH8 annotation? Also, the branch length between the algal and related and newly “discovered” fungal sequences are so short that I have more questions about the relationship of the putative algal GH8 to those in fungi than to anything in prokaryotes. To what extent have the authors explored alternative hypotheses here?

Please find our response regarding the GH8 annotation in the response to comment #4 below.

Regarding the origin/phylogeny of the gene belonging to the GH8 family the proposed conclusion is based on three layers of analyses. Please note that all three analyses are not sensitive to the annotation of the gene (the fact that it belongs to the GH8 family has no impact on the reconstructions).

- i) Comparative phylogenomics using OrthoFinder on the Chlorophytes exclusively, which resolved an orthogroup with mostly LFA species.
- ii) We then explored in more details the distribution of the genes belonging to this orthogroup by exploring by BLAST much larger databases that includes representatives of all sequenced eukaryotic and prokaryotic lineages (see details in the edited material and method). All the sequences captured by this additional survey are included in the tree presented in Figure 4a (and Figure S6 for a high-resolution image of the phylogenetic tree).
- iii) The scaffolds including the putative GH8 members in algae were re-analyzed to confirm that they are not contaminations.

Given the distribution in a limited set of bacteria and only a few classes of fungi and a single class of Chlorophyte algae we can propose two hypotheses.

Hypothesis 1: The gene family was present in the most recent common ancestor of the prokaryotes+eukaryotes (or at least the MRCA of the eukaryotes) and independent gene losses explain the distribution in extant species. However, this hypothesis would imply many independent losses across the eukaryote lineages [common ancestor of the Streptophytes, all chlorophyte lineages other than LFA Trebouxiophyceae (Mamiellophyceae, Ulvophyceae, Chlorophyceae...), Rhodophyta, Glaucophyta, SAR, Haptista, Amoebozoa, Excavates, non-fungal Opisthokontes and in various fungal lineages such as Chytridiomycota, Zoopagomycota and Dikarya] together with selective retentions (in Mucoromycotina and Trebouxiophyceae).

Hypothesis 2: Horizontal gene transfer. Two options are possible for the HGT events.

Together with the alternative scenario (Hyp 1), this is now explicitly described in the manuscript (lines 275-279):

“Such a distribution could be explained by the presence of the GH8 clade in the eukaryote most recent common ancestor, losses and its specific retention in only two clades. However, such pattern would require losses in multiple eukaryotic lineages. The other hypothesis, HGT, is more parsimonious, requiring only two events. The phylogenetic analysis thus supports at least two HGT events in the evolutionary history of the GH8 family. The GH8 originated in bacteria and was horizontally transferred to fungi and Trebouxiophyceae independently or, alternatively, the GH8 was first transferred to Mucoromycotina fungi as an intermediate recipient between bacteria and the algae (Figure 4A, Figure S6). Following this HGT, the GH8 was retained in most LFA Trebouxiophyceae species (11/13) but lost in most species that did not maintain the ability to lichenize (5/23 nLFA Trebouxiophyceae).”

4) On a related note, if I am reading your results correctly you report cleavage of both beta-1,3 and beta-1,4 linkages but GH8s cleave only beta 1,4 linkages (including beta-1,4 linkages in mixed linkage molecules). Have you explored a relationship to other CAZy families involving mixed beta 1,3/1,4 linkages, such as GH131 (in which both are cleaved, and which occur in fungi)? How sure are you this is a GH8?

In addition to covering this comment, we also combined here responses to comment #2 and #3.

The classification of the candidate orthogroup NO.HOG0012965 as part of the GH8 family was based on two complementary annotations: i) an InterProScan (IPR) annotation assigning to all the proteins belonging to that orthogroup the IPR002037 (Glycoside hydrolase, family 8, <https://www.ebi.ac.uk/interpro/entry/InterPro/IPR002037/>), ii) alignment and identification of the active sites (SuppFig S5).

To improve the reliability of the classification, we have now run a classification of CAZymes using CUPP (10.1186/s13068-019-1436-5, See new Table S10). In the revision process we have also informally discussed the CAZyme classification with the curators (Elodie Drula and Bernard Henrissat) from the CAZy team, who validated our annotation.

Please note that none of the annotation tools (nor the BLAST search) identified the sequences as GH131, or being phylogenetically-related to GH131. It is indeed known that different CAZy families can share similar substrate specificities. In the GH8 family, various enzyme activities have been reported in the literature: chitosanase (EC 3.2.1.132), endo-beta-1,4-glucanase (EC 3.2.1.4) and lichenase / endo-beta-1,3-1,4-glucanase (EC 3.2.1.73).

It would be extremely relevant in the future to study the diversity of GH families present within the Chlorophyte, lichenizing or not.

5) The orthogroup analysis identified many gene models but almost all the attention is focused on one gene coding for a putative GH8 endoglucanase. A glutathione S transferase is briefly mentioned, which is interesting in light of the key and known role glutathione regulation plays in lichen symbiosis (PNAS 102: 3141–3146). In general, I would expect the orthogroups to be much more thoroughly explored and discussed, especially in also in the context of at least some known pathways and signaling cascades. What % of the orthogroup gene models had annotations?

Among the 32 candidate orthogroups selected following the comparative phylogenomics + phylogeny + differential gene expression analysis, eight have a functional InterProScan annotation. Following the reviewer advice, we have now described these annotations in Figure 3 and, for the most interesting ones, directly in the manuscript (in the “result” (lines 233-242) and in the “discussion” (lines 321-332 & 389-395) sections).

The potential function during lichenization is proposed for three of them: the GH8 (as previously done, but with an extended discussion, as suggested in the specific comments of the reviewer), the GST and the SDR.

6) The paper is logically and sequentially structured, but I was surprised how vague the language is in key places, repeatedly stating that it will address our “elusive understanding” of lichen algae, but not clearly outlining the study objectives, questions, or hypotheses around the study system. I am missing crisply formulated language addressing known, published hypotheses around lichen algae and a review of their postulated role in and benefit from lichen symbiosis, and gaps the authors hope to fill with the present study.

We have edited the introduction (lines 99-101) to better define our objectives, and we have added details in the “results” section.

The introduction and especially the first paragraph of the discussion reach quite far into other systems for analogs, but the substantial literature on trebouxiophyte algae (1950s to the present) is barely discussed.

In this manuscript, we intend to focus on the evolution of lichenization, and the genetic gains associated with this innovation. Following the reviewer comments, we have added several connections with lichen physiology and biology (in the context of Trebouxiophyceae) in the result and the discussion section. However, as this manuscript also targets scientists interested by symbiosis and/or evolutionary processes in general we did not delve too deeply into the specifics of the trebouxiophyte algae in the introduction. Nevertheless, we have added a paragraph in the introduction to distinctly outline the knowledge gap pertaining to the genomic composition of lichen-forming algae (lines 89-98):

“The initiation of contact between lichen symbionts hinges on mutual recognition, with emerging evidence suggesting the involvement of elicitors that interact with the cell wall (reviewed in ²⁷). On the mycobiont side, fungal stimuli may encompass the activities of carbohydrate-active enzymes (CAZymes), potentially enhancing the permeability of algal cell walls ²⁸. Sugars, sugar alcohols, along with other compound groups like secondary metabolites and antioxidants, are proposed as key elements in maintaining the intricately balanced symbiotic interplay between fungi and algae in lichens ^{27,29}. This process, which exclusively involves compatible algal species, implies that LFA should manifest distinct genomic features compared to algae unable to establish symbiotic associations. In this case as well, the limited availability of genomic information for LFA has thus far impeded the testing of this hypothesis.”

For instance, the authors briefly speculate that a GH8 enzyme could be useful in acquiring carbon from sources other than photosynthesis, but do not review the literature in which trebouxioid mixotrophy has been observed and/or it has been speculated (e.g. by Ahmadjian 2002, book chapter in Seckbach, ed. Symbiosis: mechanisms and model systems. Cellular origin and life in extreme habitats) that trebouxioid algae derive carbon heterotrophically from fungi. This would seem to be quite relevant to their findings.

We completely overlooked this part of the original literature, and we thank the reviewer for pointing it out. This is a very interesting point raised by the reviewer that we now include in the discussion. We allowed ourselves to even speculate on these previous observations in the integration of the different “lichenization genes” we shortlisted (lines 383-395).

“Similarly, we can speculate that LFA engulfed in the macrolichens could generate simple sugars, in particular glucose monomers, from lichenans and use them as a carbon source as an alternative pathway to photosynthesis for carbon assimilation or as a carbon source for other microbial members of the lichen community. The direct uptake of glucose by *Trebouxia* during symbiosis has been previously reported and proposed as an additional source of carbon to increase ribitol efflux (Tapper et al. 1981). As a working model, we propose that the combined gain of the GH8, increasing carbon availability to the LFA, and expansion of the SDR family, increasing ribitol biosynthesis, contributed to the evolution of lichenization in the Trebouxiophyceae.”

7) I found the frequency with which imprecise language is used adds up to an impediment to reading the paper. The authors talk about "transcriptomic state" instead of differential expression, and "lichen-forming", etc. when this study does not address morphogenesis and there is no new or existing evidence about any role the microbes play in lichen formation. Other examples in the specific comments.

We have edited the whole manuscript to reduce as much as possible the use of imprecise language. For instance, "transcriptomic state" has been replaced by "differential gene expression" when needed. The changes can be followed in the manuscript with track changes.

The term "lichen-forming" requires a specific response as it is also mentioned in the specific comments of the reviewer below.

Quote from the reviewer specific comment: "Line 52: why lichen-forming algae? There is no evidence that the algae form lichens, so perhaps something simpler, like lichen algae, or algal photobionts, would be appropriate"

Several terms have been used in the literature to define algal species that are known as algal photobionts in lichens, including "Lichen-Forming Algae" (*i.e.* Nelsen et al. 2020). We decided to use this terminology, and not for instance Algal Photobiont, as it mirrors the commonly used "Lichen-Forming Fungi" (*i.e.* . Nelsen et al. 2020; Wang *et al.* 2023). The LFA refer here to algae actually isolated from lichens.

To avoid any confusion, we define what we mean by "Lichen-Forming Algae" when first introducing it (lines 77-78):

"On the photobiont side, LFA algal species which are known to establish the lichen symbiosis (thereafter called Lichen-Forming Algae or LFA) are almost exclusively found in two of the eleven chlorophyte algae classes, the Ulvophyceae and the Trebouxiophyceae"

In summary, I think there are solid analyses at the core of this study with some genuinely novel and exciting results but I feel like the authors do not sufficiently relate these to existing questions on lichen symbiosis and focus on one gene while only shallowly exploring the rest of their data. I especially hope that the authors can show that they have explored a wider range of possibilities regarding the origin and even annotation of the putative GH8.

We have carefully taken into account the major and specific comments provided by the reviewer and express our gratitude for the highly constructive review.

Specific comments:

Abstract: first opening sentences ("our understanding of their evolution and function remains elusive") are way too vague and do not really set up the question behind the study

It is important for us to bring this work in a wider picture which is the general interest of our lab, the evolution of mutualistic interactions. We believe this study fits in this larger scope as what we learnt

from lichen evolution provides knowledge to build a global picture on these evolutionary processes. This is also why part of the discussion focusses on more general evolutionary concepts. Although lichens are at the heart of this work, the evolution of mutualism aspect is core to this work.

We have rephrased the abstract to clearly separate these two aspects. The opening sentence of the abstract is now (lines 28-31):

“Mutualistic symbioses have contributed to major transitions in the evolution of life and are at the center of extant ecosystems. Here, we investigated the evolutionary history and the molecular innovations at the origin of one of them, the lichens, which is formed between fungi and green algae or cyanobacteria.”

Independent gains of symbiosis in Trebouxiophyceae and Ulvophyceae really new here?

We did not intend to present this as a novelty. The approach confirming this is novel and supports the consensus hypothesis for the evolution of the trait. We have rephrased the sentence (line 36):

“We identified at least three independent gains of the ability to form lichens, one in Trebouxiophyceae and two in Ulvophyceae, confirming the convergent evolution of the lichen symbioses.”

Abstract, line 38: GH8 is a family of GHs and not one specific enzyme, as implied here

We corrected the sentence to reflect this.

Line 52: either

We corrected the typo

Line 52: why lichen-forming algae? There is no evidence that the algae form lichens, so perhaps something simpler, like lichen algae, or algal photobionts, would be appropriate

Please refer to our reply to major comment #7 explaining why we decided to use Lichen-Forming Algae, one option among several terms used in the literature.

Line 61: what does “successfully unraveled the evolution” mean?

We have edited the sentence for clarity (lines 66-70):

“The comparison of genomes in a defined phylogenetic context (comparative phylogenomics) has successfully unraveled the evolutionary history of several mutualistic symbioses with complex evolutionary patterns, combining gains and losses across lineages.”

Line 65: why lichen-forming fungi? Again, neither partner is known to be able to “form” lichens, and this paper is not about that, so “lichen fungi” or “lichen fungal symbionts” might be more appropriate

We fully agree that several terms would be valid here. We picked the term Lichen-forming fungi (LFF) as it is used in the literature, among other terms, to define fungal species “able to engage into the lichen symbiosis”. The term was used in a recent study published in Nature Communications (Wang et al. 2023), so we assume it is appropriate for this journal.

Line 66: lichen studies

This sentence has been edited.

Line 67: what was studied is fungal involvement in lichens, not the ability to “form” them, which has not been studied (morphogenesis)

We have used this term as it is used in the literature.

Line 71-72: again, this study is not about lichen-*** forming *** ability, which implies morphogenesis. It simply addresses occurrence in lichens

This comment has been addressed above in the response to major comment #7.

Line 71-73: who has ever proposed that the common ancestor of Trebouxiophyceae and Ulvophyceae was involved in lichens? I am not aware anyone has, and none of the citations provided (12,13,22,2) support this statement. It would be better to acknowledge that this is being proposed as a hypothesis de novo in this manuscript, but I have my doubts whether there are really reasonable grounds to hypothesize this.

We have taken this comment into account. Please see our response to major comment #1.

Line 77: the English is not clear (“constrained so far analyses”)

We corrected the sentence (lines 85-87):

“However, the limited availability of LFA genomes has so far constrained molecular analyses to single algal species such as *Asterochloris glomerata* and *Trebouxia sp.* TZW2008”.

Line 79: “the evolutionary history of lichens on the green algal side and the underlying molecular mechanisms remains elusive” — is the objective of this paper to redress this problem? The objectives and hypotheses should be clearly stated

The objectives of the study are now stated in the introduction (lines 99-101):

“In this study, we deployed unsupervised phylogenomic comparative approaches to decipher the evolutionary history and the genetic mechanisms conferring to certain chlorophytes species the ability to engage into the lichen symbiosis.”

Line 81: I can deduce what nLFA stands for but it is never explained

Thank you for noticing this mistake. We added the meaning of nLFA (non Lichen-Forming Algae, lines 102-103).

Line 82: not clear what is meant by “129 previously reported genomes (n = 26)”; I see that the 129 is the sum of 26+103 but perhaps this could be worded as “129 genomic datasets / species comprised of 26 genomes and 103 transcriptomes”. Also, these numbers do not agree with those given on lines 108-109 (35+106 = 141).

We modify line 82 according to the reviewer suggestion: “with 26 genomes and 103 transcriptomes of chlorophyte algae publicly available” (lines 104-105).

On line 131, the 141 number correspond to the combination of publicly available genomes (26) and transcriptomes (103) with genomes and transcriptomes sequenced in this study (9 genomes and 3 transcriptomes): $26 + 9 + 103 + 3 = 141$.

We also modified the Figure 1B to make the database content more understandable.

Line 119: surely some steps are being left out here, as OrthoFinder does not generate phylogenetic trees.

OrthoFinder does generate a species tree based on the orthogroups shared across species using the STAG method (see <https://github.com/davidemms/OrthoFinder#species-tree-directory>).

Line 121: what does it mean that “symbiotic abilities were defined based on 21 and finally curated by the authors of this paper.” Perhaps better “the status of algal species as symbionts was assigned following Sanders & Matsumoto²¹” — please explain what the additional curation means

We rephrased the sentence (lines 150-155): “The status of algal species as symbionts was assigned following on ²¹. Furthermore, given that many green algal species lack distinct morphological features and their lichenization status has often only been reported in light microscopic studies, the determination of the lichenization status for each species in this study was based on a thorough review of published studies based on sequence data.”

Lines 131-136: It has previously been observed that lichen symbioses are distinct phenomena polyphyletic in both the fungal and algal phylogenies (e.g. New Phytologist 234: 1566-1582). However, I fail to see how the evidence acquired in the present ASR supports the interpretation that “molecular mechanisms ... likely differ”. The authors did not study this.

This sentence has been removed as it lacked clarity.

Lines 139-140: first sentence can be deleted

We edited the paragraph accordingly and integrated the first sentence in a better way (lines 172-175):

“Our gathered dataset encompasses 13 LFA and 23 nLFA in the Trebouxiophyceae class allowing to conduct comparative genomics to identify genomic or genetic features linked to the gain of lichenization/lichen-forming abilities.”

Line 161-163: exactly, this should be clearly stated as a caveat throughout As mentioned above (in response to major comment #2), we now clearly explain the criteria we use to define genes as “lichenization genes”, and two sentences have been added in the result (lines 227-231)

Line 164: “associated” is better than “linked”

We modified all instances in the text accordingly.

Line 172: you state that you “recomputed” the DE genes, but in Table S8 the data are indicated as being from Kono et al.. Is this a new analysis?

We downloaded the raw reads from Kono et al. and recomputed the differentially expressed genes after mapping on the newly annotated genome. This re-analysis of the RNAseq was necessary to be able to cross-reference the comparative phylogenomics and the differential gene expression analysis. The analysis is new, but the reads are the ones coming from the Kono et al. study.

Line 194: please replace “glycosyl hydrolase of the eighth family” with a more standard explanation that GHs are classified in the CAZy database as families and this is GH family 8. Also please get someone to revise this manuscript who speaks English fluently

We have rephrased according to the reviewer suggestion (lines 236-237):

“Additionally, some candidates are LFA-specific, as seen in NO.HOG0012965, which contains a carbohydrate-active enzyme belonging to the Glycoside Hydrolase family 8.”

Line 253-254: this doesn't belong in the results

The sentence was removed from the results section.

Line 261: please don't call GH8 a gene, it is a CAZy family

We corrected the sentence (lines 341-343):

“The other horizontally-acquired orthogroup is the one that best discriminates LFA Trebouxiophyceae from other Chlorophytes at the phylogenetic level and is annotated as belonging to the GH8 family”

Line 263: cycads

We meant the genus *Cycas* but forgot the italic font. It is now corrected.

Line 268: bark (can't be made plural)

We corrected this error.

L256-274: it is good to place these findings in context but this paragraph becomes rambling and goes off topic, addressing issues such as potentiation etc. that have no relevance to what has been shown with this study

We understand the reviewer point of view however, as mentioned earlier, we believe our study is not only relevant in a “lichen” context, but also in the more general context of the evolution of symbiotic interactions. The macroevolutionary events leading to such associations to evolve are very often unknown. Our study does provide an original view on this process in lichen, that can be extrapolated – conceptually – to other mutualistic associations. As this is a section of the discussion, we decided to maintain it.

We also realize that our discussion was not integrating enough our findings in the context of lichen biology and physiology. We have added the description of additional genes strongly associated with lichenization and their putative roles in lichens. We find this revisited discussion more balanced and hope the reviewer will find it more relevant.

L288: provide a citation for the fungus possessing enzymes to break down algal cell walls
We now cite Kono et al. 2020 (<https://doi.org/10.1186/s12864-020-07086-9>) and Resl et al. 2021 (<https://doi.org/10.1038/s41467-022-30218-6>). These two papers nicely cover the potential for the Lichen-Forming Fungi to degrade cell-wall polysaccharides (lines 373-374).

L296ff.: “we can speculate that LFA engulfed in the macrolichens could generate simple sugars from lichenans and use them as a carbon source as an alternative pathway to photosynthesis for carbon assimilation” — there is a fair amount of experimental literature on Trebouxia using external carbon sources, this would be good to tie into here with some citations

This is an excellent suggestion. We had completely ignored this literature. We now cite one study and integrate this process in our working model (lines 388-391).

L371: please disclose how many of the cultures were axenic versus containing some bacteria or fungi

We have clarified the information. Even if cultures were not axenic in the sense that we did not test with bacterial primers, algae were cultivated for months and transferred multiple times without any trace of contamination. In addition, as indicated in the manuscript (paragraph *Genome assembly* in the Material & Methods), only scaffolds assigned to the chlorophytes via MEGAN were kept for the downstream analyses.

L468: what does “crossing” mean here?

We cross-referenced the gene IDs present in the HOG coming from the comparative phylogenomics with the list of gene IDs from *Trebouxia* sp TZW2008 found differentially regulated during symbiosis (either in co-culture or in natural lichens) according to the (reanalysed) data from Kono et al (2020).

This is what we describe in the result section and does not belong to the Material and Method section. We have removed this sentence.

L504: “the protein *A. glomerata*” — something is wrong or missing here, *A. glomerata* is one of your species

Thanks for noticing this, we modified the sentence as follow: “We selected the protein from *A. glomerata*”

L563: authors?

Thanks for noticing this, we added the author name:

“ Nash, T. H. *Lichen Biology*. (Cambridge University Press, Cambridge ; New York, 2008)”

Page 29: Figure 2A is unreadable

We provide new figures that have been improved for readability.

Page 30: I am confused by the plotting of orthogroups and results of differential expression analysis (called here “transcriptomic state”). If I understand correctly, the DE analysis is in fact a reanalysis of the Kono et al. data for just one of the black-dotted *Trebouxia* strains, right? Am I to understand that each orthogroup numbered n0.hog... contains/includes at least one gene model with log fold change differences (?) indicated on the unlabeled scale bottom right? I am especially confused how you juxtapose differential expression statistics which presumably refer to a single gene model in one strain next to mapped orthogroups which contain many genes. This is at a minimum very confusing and unreadable in its present form, and if it is plotted the way I think it is it is also misleading.

The orthogroups plotted in Figure 3 are indeed the orthogroups that contain at least one gene differentially expressed in one of the symbiosis states (in the co-culture and/or in field-collected lichens) in the *Trebouxia* strain analysed in Kono et al, 2020. We apologize if the way we presented the data was unreadable.

The legend of the figure was modified to improve clarity and avoid any confusion:

“Figure 3: Chlorophytes phylogeny, heatmap of the number of genes per species and per orthogroup for the ones that contain at least one differentially expressed gene (up and/or down-regulated) in symbiosis for *Trebouxia* sp (TZW2008) according to the data from Kono et al, 2020, the contribution of each orthogroup according to Figure 2, the transcriptomic state of the differentially expressed genes in symbiosis and the main functional annotation of each orthogroup (all the IPR domains and GO terms found in the orthogroups are available in Table S7) . Lichen forming Trebouxiophyceae are indicated with black dots. The orthogroups with no functional annotation have an unknown function.”

Page 31: I am not seeing the black dots or green line the authors refer to

Thank you for noticing. The bars are orange on the panel 4B, we corrected the legend accordingly.

The black dots were at the tips of the phylogenetic tree in panel 4A. As almost all the sequences contain the GH8 functional domain, we removed the black dots from the figure and added the information in Table S11.

Reviewer #2 (Remarks to the Author):

Here, the authors provide novel and critical insight into lichen evolution from the perspective of the photobiont. Their findings of the combined roles gene family expansion and horizontal gene transfer are particularly insightful. This work provides valuable, original insight into evolutionary genomics in one of the most iconic symbiotic systems. Data collection, analyses and interpretations are appropriate; and the claims and conclusions are supported by the data. Any the methodology is described in enough detail for work to be reproduced. I recommend the study for publication.

We warmly thank the reviewer for this very positive feedback. The submitted manuscript was the results of years of work, and we believe the edited version is now even improved.

REVIEWERS' COMMENTS

Reviewer #1 (Remarks to the Author):

The manuscript is much improved and the authors have done an excellent job addressing almost all of the major comments from the first round. I have given the paper another read and suggest a few mostly minor edits (see below). My only remaining major concern relates to the authors' continued misuse of the term 'lichen-forming' for both fungi and algae, which along with some other imprecise language I criticized during the first round. The authors' response was:

"Several terms have been used in the literature to define algal species that are known as algal photobionts in lichens, including "Lichen-Forming Algae" (i.e. Nelsen et al. 2020). We decided to use this terminology, and not for instance Algal Photobiont, as it mirrors the commonly used "Lichen-Forming Fungi" (i.e. . Nelsen et al. 2020; Wang et al. 2023). The LFA refer here to algae actually isolated from lichens.

To avoid any confusion, we define what we mean by "Lichen-Forming Algae" when first introducing it (lines 77-78):

"On the photobiont side, LFA algal species which are known to establish the lichen symbiosis (thereafter called Lichen-Forming Algae or LFA) are almost exclusively found in two of the eleven chlorophyte algae classes, the Ulvophyceae and the Trebouxiophyceae" "

It is true that this usage has precedent, but it is also true that literature on lichen symbiosis is riddled with imprecise, inaccurate, and outright misleading use of terminology, which sometimes makes it very difficult to advance the field. For instance, what if someone actually wants to study lichen formation and finds that some lichen algae contribute to morphogenesis and others don't? Perhaps these algae are lichen-forming and the others are not? The same is true for the fungus. Do we really know that the fungus is 'lichen-forming', or is the thallus perhaps an emergent property resulting from stable multi-player input? We don't know, and your study is not about this, but by using this language you are perpetuating poor practices and making it difficult for people who actually study this to talk about untested hypotheses, because the "fact" that fungi and algae "form" lichens is already baked into your language. The previous use of this term in the literature is no justification for its continued unreflected use. Instead, its needless appearance in this manuscript threatens to be an example of snowballing bias / use of imprecise terminology in science.

There are many phrases that can replace 'lichen-forming' that do not assert something you do not know. The classical way is photobiont. Another way that has started to gain traction is to refer to them as lichen fungal or algal symbionts, and in sentences as 'involved in lichens'. The authors almost accurately described this "trait" in lines 156-157 (chlorophyte alga inside a fungal thallus) without using biased language.

I also urge the authors to be more precise with related terms such as "the ability to lichenize" (L168), lichenization (L215 and several other places), etc. You know these algae have a lifestyle inside of lichens, and say so. It is not clear what 'lichenize' really means (and precedent isn't helpful here either). I think what you really mean is occurrence in, a lifestyle in, lichens/lichen symbioses. I am not sure how a GH become a lichenization gene — what does that mean, exactly? Is every orthogroup that appears in association with lichen symbiosis, in every microbe, automatically a lichenization gene? Keep it factual and use tight definitions, it will pay off in the long run.

I think adjusting the language is an easy fix and would really urge the authors to consider this in the interests of contributing to cleaning up lichen science. Otherwise it's really a great paper — I look forward to seeing it published.

Minor comments:

L58: "are likely important for its biology" would be more accurate, as this hasn't been proven

L88: not clear what 'this process' refers to. I think what the authors want to say is that symbiosis can be expected to shape genome architecture and content and this is to be expected for lichen algae — plenty of references they could cite here e.g. from Nancy Moran on bacteria, Francis Martin/Annegret Kohler on fungi, etc.

L88: 'which exclusively involves compatible algal species' — of course it does, nobody expects incompatible algae to be involved. Can be removed.

L93-94: conferring to certain chlorophytE species ... engage IN lichen symbiosis

L115: fuscideae

L134: I would not say it has been ignored, perhaps it would be more accurate to say "has not been explored"

L200: HOG not used anywhere else or defined

L215: please do not call these lichenization genes. As mentioned above, 'lichenization' is an ill-defined concept. Please use something more neutral.

L280,281: they asserts something you do not know and is unrelated to the present study

L298: reductase

L299: 'acyclic' is probably more accurate than 'linear'

L300: aphthosa

L301: current name is *Gyalolechia bracteata*

L309: a glutathione-S-transferase. Don't capitalise the enzyme names and pay attention to singular versus plural — there are still too many small mistakes like this and I have highlighted only a few here. One round of proofreading from somebody who speaks English as their first language would still really help.

L361-364: exactly — this 'working model' is the correct way to present your hypotheses about the role of GH8 and the genes coding for SDR family enzymes in the evolution of algal involvement in lichen symbioses. However, I do not understand though how the authors can go from this to 'lichenization genes' three lines later (L367) and in other places of the manuscript.

L769: This is the Seckbach book. Please cite the specific chapter by Ahmadjian, with Ahmadjian as the given author, and the chapter as pp. xxx-xxx in the book (editors), title, in normal Nature Communications format. Check their style guide for citing a chapter in an edited book.

L965: please do not call these lichenization genes, see comment in main section

L990: as follows; chlorophyte algae

RESPONSE TO REVIEWERS' COMMENTS

Reviewer #1 (Remarks to the Author):

The manuscript is much improved and the authors have done an excellent job addressing almost all of the major comments from the first round. I have given the paper another read and suggest a few mostly minor edits (see below). My only remaining major concern relates to the authors' continued misuse of the term 'lichen-forming' for both fungi and algae, which along with some other imprecise language I criticized during the first round. The authors' response was:

"Several terms have been used in the literature to define algal species that are known as algal photobionts in lichens, including "Lichen-Forming Algae" (i.e. Nelsen et al. 2020). We decided to use this terminology, and not for instance Algal Photobiont, as it mirrors the commonly used "Lichen-Forming Fungi" (i.e. . Nelsen et al. 2020; Wang et al. 2023). The LFA refer here to algae actually isolated from lichens.

To avoid any confusion, we define what we mean by "Lichen-Forming Algae" when first introducing it (lines 77-78):

"On the photobiont side, LFA algal species which are known to establish the lichen symbiosis (thereafter called Lichen-Forming Algae or LFA) are almost exclusively found in two of the eleven chlorophyte algae classes, the Ulvophyceae and the Trebouxiophyceae" "

It is true that this usage has precedent, but it is also true that literature on lichen symbiosis is riddled with imprecise, inaccurate, and outright misleading use of terminology, which sometimes makes it very difficult to advance the field. For instance, what if someone actually wants to study lichen formation and finds that some lichen algae contribute to morphogenesis and others don't? Perhaps these algae are lichen-forming and the others are not? The same is true for the fungus. Do we really know that the fungus is 'lichen-forming', or is the thallus perhaps an emergent property resulting from stable multi-player input? We don't know, and your study is not about this, but by using this language you are perpetuating poor practices and making it difficult for people who actually study this to talk about untested hypotheses, because the "fact" that fungi and algae "form" lichens is already baked into your language. The previous use of this term in the literature is no justification for its continued unreflected use. Instead, its needless appearance in this manuscript threatens to be an example of snowballing bias / use of imprecise terminology in science.

There are many phrases that can replace 'lichen-forming' that do not assert something you do not know. The classical way is photobiont. Another way that has started to gain traction is to refer to them as lichen fungal or algal symbionts, and in sentences as 'involved in lichens'. The authors almost accurately described this "trait" in lines 156-157 (chlorophyte alga inside a fungal thallus) without using biased language.

I also urge the authors to be more precise with related terms such as "the ability to lichenize" (L168), lichenization (L215 and several other places), etc. You know these algae have a lifestyle inside of lichens, and say so. It is not clear what 'lichenize' really means (and precedent isn't helpful here either). I think what you really mean is occurrence in, a lifestyle in, lichens/lichen symbioses. I am not sure how a GH become a lichenization gene – what does that mean, exactly? Is every orthogroup that appears in association with lichen symbiosis, in every microbe, automatically a lichenization gene? Keep it factual and use tight definitions, it will pay off in the long run.

We have defined lichenization as proposed by the reviewer at first mention.

I think adjusting the language is an easy fix and would really urge the authors to consider this in the interests of contributing to cleaning up lichen science. Otherwise it's really a great paper – I look forward to seeing it published.

Minor comments:

L58: "are likely important for its biology" would be more accurate, as this hasn't been proven

Done

L88: not clear what 'this process' refers to. I think what the authors want to say is that symbiosis can be expected to shape genome architecture and content and this is to be expected for lichen algae – plenty of references they could cite here e.g. from Nancy Moran on bacteria, Francis Martin/Annegret Kohler on fungi, etc.

Done

L88: 'which exclusively involves compatible algal species' – of course it does, nobody expects incompatible algae to be involved. Can be removed.

Done

L93-94: conferring to certain chlorophytE species ... engage IN lichen symbiosis

Done

L115: fuscideae

Done

L134: I would not say it has been ignored, perhaps it would be more accurate to say "has not been explored"

Done

L200: HOG not used anywhere else or defined

Now defined line 159 in the title

L215: please do not call these lichenization genes. As mentioned above, 'lichenization' is an ill-defined concept. Please use something more neutral.

Removed and replaced

"Reverse genetic analyses will be required in the future to validate their functions when a genetically tractable system and the *in vitro* resynthesis or lichen formation will have been developed. Among the candidate genes associated with the symbiosis"

L280,281: they asserts something you do not know and is unrelated to the present study

This comment refers to the following sentence “These results suggest that the GH8 enzyme was acquired through horizontal gene transfer (HGT) in the MRCA of Trebouxiophyceae, along with the ability to interact with LFS. The enzyme was later retained in species that engage in symbiosis. ”

We actually demonstrate that the GH8 has a GH8 activity on lichenan, and that it comes from HGT (according to the phylogeny and genomics) concomitantly with the ability to engage in symbiosis according to the ASR. We are not sure to understand the reviewer comment.

L298: reductase

Done

L299: ‘acyclic’ is probably more accurate than ‘linear’

Done

L300: aphthosa

Done

L301: current name is Gyalolechia bracteate

Done

L309: a glutathione-S-transferase.

Done

Don’t capitalise the enzyme names and pay attention to singular versus plural – there are still too many small mistakes like this and I have highlighted only a few here. One round of proofreading from somebody who speaks English as their first language would still really help.

Done. The manuscript has been edited by a native English speaker.

L361-364: exactly – this ‘working model’ is the correct way to present your hypotheses about the role of GH8 and the genes coding for SDR family enzymes in the evolution of algal involvement in lichen symbioses. However, I do not understand though how the authors can go from this to ‘lichenization genes’ three lines later (L367) and in other places of the manuscript.

Changed (to “genes associated with the symbiosis/lichenization”)

L769: This is the Seckbach book. Please cite the specific chapter by Ahmadjian, with Ahmadjian as the given author, and the chapter as pp. xxx-xxx in the book (editors), title, in normal Nature Communications format. Check their style guide for citing a chapter in an edited book.

Done

L965: please do not call these lichenization genes, see comment in main section

Done

L990: as follows; chlorophyte algae

Done